# MSH2 shapes the meiotic crossover landscape in relation to interhomolog polymorphism in Arabidopsis

Alexander R Blackwell[1,†] (iD), Julia Dluzewska[2,†] (iD), Maja Szymanska-Lejman[2] (iD), Stuart Desjardins[3] (iD), Andrew J Tock[1] (iD), Nadia Kbiri[2] (iD), Christophe Lambing[1] (iD), Emma J Lawrence[1], Tomasz Bieluszewski[2], Beth Rowan[4,‡] (iD), James D Higgins[3], Piotr A Ziolkowski[2,*] (iD) & Ian R Henderson[1,**] (iD)

## Abstract

During meiosis, DNA double-strand breaks undergo interhomolog repair to yield crossovers between homologous chromosomes. To investigate how interhomolog sequence polymorphism affects crossovers, we sequenced multiple recombinant populations of the model plant *Arabidopsis thaliana*. Crossovers were elevated in the diverse pericentromeric regions, showing a local preference for polymorphic regions. We provide evidence that crossover association with elevated diversity is mediated via the Class I crossover formation pathway, although very high levels of diversity suppress crossovers. Interhomolog polymorphism causes mismatches in recombining molecules, which can be detected by MutS homolog (MSH) mismatch repair protein heterodimers. Therefore, we mapped crossovers in a *msh2* mutant, defective in mismatch recognition, using multiple hybrid backgrounds. Although total crossover numbers were unchanged in *msh2* mutants, recombination was remodelled from the diverse pericentromeres towards the less-polymorphic sub-telomeric regions. Juxtaposition of megabase heterozygous and homozygous regions causes crossover remodelling towards the heterozygous regions in wild type *Arabidopsis*, but not in *msh2* mutants. Immunostaining showed that MSH2 protein accumulates on meiotic chromosomes during prophase I, consistent with MSH2 regulating meiotic recombination. Our results reveal a pro-crossover role for MSH2 in regions of higher sequence diversity in *A. thaliana*.

**Keywords** *Arabidopsis*; crossover; meiosis; MSH2; polymorphism
**Subject Categories** Cell Cycle; DNA Replication, Recombination & Repair; Plant Biology
**The EMBO Journal (2020) 39: e104858**

## Introduction

Meiosis creates genetic diversity during eukaryotic sexual reproduction and is widely conserved among plants, animals and fungi (Villeneuve & Hillers, 2001; Mercier *et al*, 2015). DNA double-strand breaks (DSBs) form during meiotic prophase I and may enter an interhomolog repair pathway to create reciprocal crossovers or non-reciprocal gene conversions (Villeneuve & Hillers, 2001; Mercier *et al*, 2015). Together, these have a profound effect on genetic variation and genome evolution. SPO11 transesterases form DSBs during meiosis, which are resected to produce 3′-overhanging single-stranded DNA (ssDNA) (Keeney & Neale, 2006; Hunter, 2015). Meiotic ssDNA is bound by the RecA homologs DMC1 and RAD51, which mediate strand invasion of a homologous chromosome to form displacement loops (D-loops) (Keeney & Neale, 2006; Hunter, 2015). D-loops may be dissolved and repaired to form a non-crossover or protected and further processed to form a crossover (Hunter, 2015). The conserved Class I "ZMM" pathway provides the major activity for crossover formation in plants (Mercier *et al*, 2015; Pyatnitskaya *et al*, 2019). The ZMM pathway acts to stabilize interhomolog joint molecules and promote crossovers via resolution of double Holliday junctions (dHJs) (Hunter, 2015; Mercier *et al*, 2015). A minority of crossovers are formed by the Class II pathway that involves structure-specific endonucleases, including MUS81 (Hunter, 2015; Mercier *et al*, 2015). An important distinction between these repair pathways is that only Class I crossovers show interference, meaning that events are more widely spaced along the chromosomes than expected by chance (Villeneuve & Hillers, 2001; Copenhaver *et al*, 2002; Hunter, 2015; Mercier *et al*, 2015).

Due to sequence polymorphism between homologous chromosomes, interhomolog joint molecules that form during meiosis are

1  Department of Plant Sciences, University of Cambridge, Cambridge, UK
2  Laboratory of Genome Biology, Institute of Molecular Biology and Biotechnology, Adam Mickiewicz University, Poznan, Poland
3  Department of Genetics and Genome Biology, University of Leicester, Leicester, UK
4  Department of Molecular Biology, Max Planck Institute for Developmental Biology, Tübingen, Germany
   *Corresponding author. Tel: +48 61 8295966; E-mail: pzio@amu.edu.pl
   **Corresponding author. Tel: +44 01223 748 977; E-mail: irh25@cam.ac.uk
   †These authors contributed equally to this work
   ‡Present address: Genome Center and Department of Plant Sciences, University of California, Davis, CA, USA

likely to experience base pair mismatches. Increased levels of sequence divergence have been observed to suppress homologous recombination (HR) during both mitosis and meiosis (Borts & Haber, 1987; Datta *et al*, 1997; Elliott *et al*, 1998; Chen & Jinks-Robertson, 1999; Opperman *et al*, 2004; Emmanuel *et al*, 2006; Li *et al*, 2006; Do & LaRocque, 2015). For example, higher levels of interhomolog divergence at a meiotic recombination hotspot in the budding yeast *Saccharomyces cerevisiae* cause decreased crossover whilst increasing the frequency of other repair events (Borts & Haber, 1987). Furthermore, structural variation within meiotic recombination hotspots, including insertions and deletions, has been associated with local crossover suppression (Dooner, 1986; Jeffreys & Neumann, 2005; Baudat & de Massy, 2007; Cole *et al*, 2010). In many cases, inhibition of HR via sequence mismatches is dependent on MutS-related heterodimers that contain MSH2 (Borts & Haber, 1987; Chen & Jinks-Robertson, 1999; Elliott & Jasin, 2001; Emmanuel *et al*, 2006; Do & LaRocque, 2015). MutS heterodimers are widely conserved complexes capable of binding sequence mismatches that play key roles in post-replicative mutation correction and rejection of heteroduplex DNA during recombination (Modrich & Lahue, 1996; Kunkel & Erie, 2005). For example, budding yeast Msh2 acts as an anti-recombinase during HR via recruitment of the Sgs1 helicase, a homolog of human BLM, which promotes disassembly of mismatched D-loops (Myung *et al*, 2001; Mazina *et al*, 2004). Hence, the mismatch repair machinery can limit HR in a polymorphism-dependent manner.

Despite the inhibitory effect of sequence divergence on HR, other data indicate a positive relationship between meiotic recombination and interhomolog polymorphism. For example, historical recombination, as measured via linkage disequilibrium, is positively correlated with sequence diversity in multiple species (Begun & Aquadro, 1992; Nordborg *et al*, 2005; Spencer *et al*, 2006; Gore *et al*, 2009; Paape *et al*, 2012; Cutter & Payseur, 2013). Direct measurements of crossover frequency in plant hybrids have also shown higher recombination compared with inbred backgrounds, in specific chromosome regions (Barth *et al*, 2001; Ziolkowski *et al*, 2015). Finally, juxtaposition of megabase regions of heterozygosity and homozygosity in *Arabidopsis* causes increased crossovers in the heterozygous regions, at the expense of the homozygous regions (Ziolkowski *et al*, 2015). In this case, remodelling of crossover frequency towards the heterozygous regions was shown to require the Class I interfering repair pathway (Ziolkowski *et al*, 2015). Therefore, the relationship between polymorphism and crossover recombination is likely to vary across different scales and chromosome contexts.

We sought to further explore the relationship between sequence polymorphism, meiotic crossover frequency and mismatch repair in the model plant *Arabidopsis thaliana*. *Arabidopsis* MSH2 forms heterodimers with MSH3, MSH6 and MSH7 *in vitro*, which display varying affinities for single base and short (1–3 bp) indel mismatches (Culligan & Hays, 2000; Adé *et al*, 2001; Wu *et al*, 2003). Single base and short indel mutation rates are elevated in *msh2* mutants, which results in deleterious phenotypes following inbreeding (Leonard *et al*, 2003; Hoffman *et al*, 2004; Watson *et al*, 2016; Belfield *et al*, 2018). Somatic HR has been measured in *Arabidopsis* using split *GUS* (*GU:US*) transgenes (Emmanuel *et al*, 2006; Li *et al*, 2006). Increasing the number of mismatches

between the *GU* and *US* substrate repeats decreases HR frequency in wild type, with higher recombination observed in *msh2* (Emmanuel *et al*, 2006; Li *et al*, 2006). Interestingly, previous work also demonstrated an increase in meiotic crossover frequency within a sub-telomeric region in *msh2* hybrids, compared with wild type (Emmanuel *et al*, 2006). Therefore, we sought to extend previous studies by mapping meiotic crossover distributions genome-wide in wild type and *msh2* and examine relationships with polymorphism.

We show that wild type crossovers are promoted in the diverse pericentromeric regions and associate with higher polymorphism at the local scale. We provide genetic evidence that crossover association with higher SNP density involves the Class I repair pathway. Due to the role of MSH2 as a mismatch sensor during HR (Borts & Haber, 1987; Chen & Jinks-Robertson, 1999; Elliott & Jasin, 2001; Emmanuel *et al*, 2006; Do & LaRocque, 2015), we sought to test its role in regulating meiotic crossover formation in *Arabidopsis*. Unexpectedly, we found that crossover association with regions of higher heterozygosity is promoted by MSH2. This leads to a re-evaluation of the impact of sequence polymorphism on meiotic recombination in plant genomes, and the role of MSH2 in this process.

# Results

### Crossover and diversity landscapes in the *Arabidopsis* genome

*Arabidopsis thaliana* predominantly self-fertilizes and occurs as naturally inbred accessions that are estimated to outcross at a rate between 0.3 and 2.5% (Bomblies *et al*, 2010; The 1001 Genomes Consortium *et al*, 2016). To explore interactions between interhomolog polymorphism and crossovers, we generated $F_2$ populations from five crosses between the Col-0 reference accession (hereafter Col) and the L*er*-0 (hereafter Ler) (Serra *et al*, 2018b), Bur-0 (hereafter Bur) (Lawrence *et al*, 2019), Ws-4 (hereafter Ws) and Ct-1 (hereafter Ct) accessions. We also generated an $F_2$ population by crossing Col and CLC, which is a mosaic of Cvi-0, Ler and Col (Fig 1). The CLC genome is composed predominantly of Cvi-0 sequence, but with a substituted Ler chromosome 5 and additional Col and Ler introgressions on the other chromosomes (Fig 1B). We also compared the Col × Ler $F_2$ population with a larger data set generated from 2,182 Col × Ler $F_2$ individuals that identified 17,077 crossovers (Serra *et al*, 2018b; Rowan *et al*, 2019). These backgrounds show a range of divergence levels when compared to Col. For example, between 413,830 (3.31 SNPs/kb) and 562,423 (4.49 SNPs/kb) SNPs were identified for each accession, relative to Col, with 30–55% of SNPs shared between Ler, Bur, Ws and Ct (Appendix Tables S1 and S2).

We sequenced genomic DNA from between 180 and 305 $F_2$ individuals from each population and identified between 1,396 and 2,478 crossovers per population, using the TIGER pipeline (Fig 1A and Appendix Table S3) (Serra *et al*, 2018b; Lawrence *et al*, 2019; Rowan *et al*, 2019). Crossovers were identified as haplotype switches along the chromosomes and were assigned between pairs of SNPs (Rowan *et al*, 2015). The SNP pairs that define the crossovers in each $F_2$ population had a mean distance in the range of 653 and 2,261 bp (Appendix Table S3). Crossover numbers per $F_2$ were

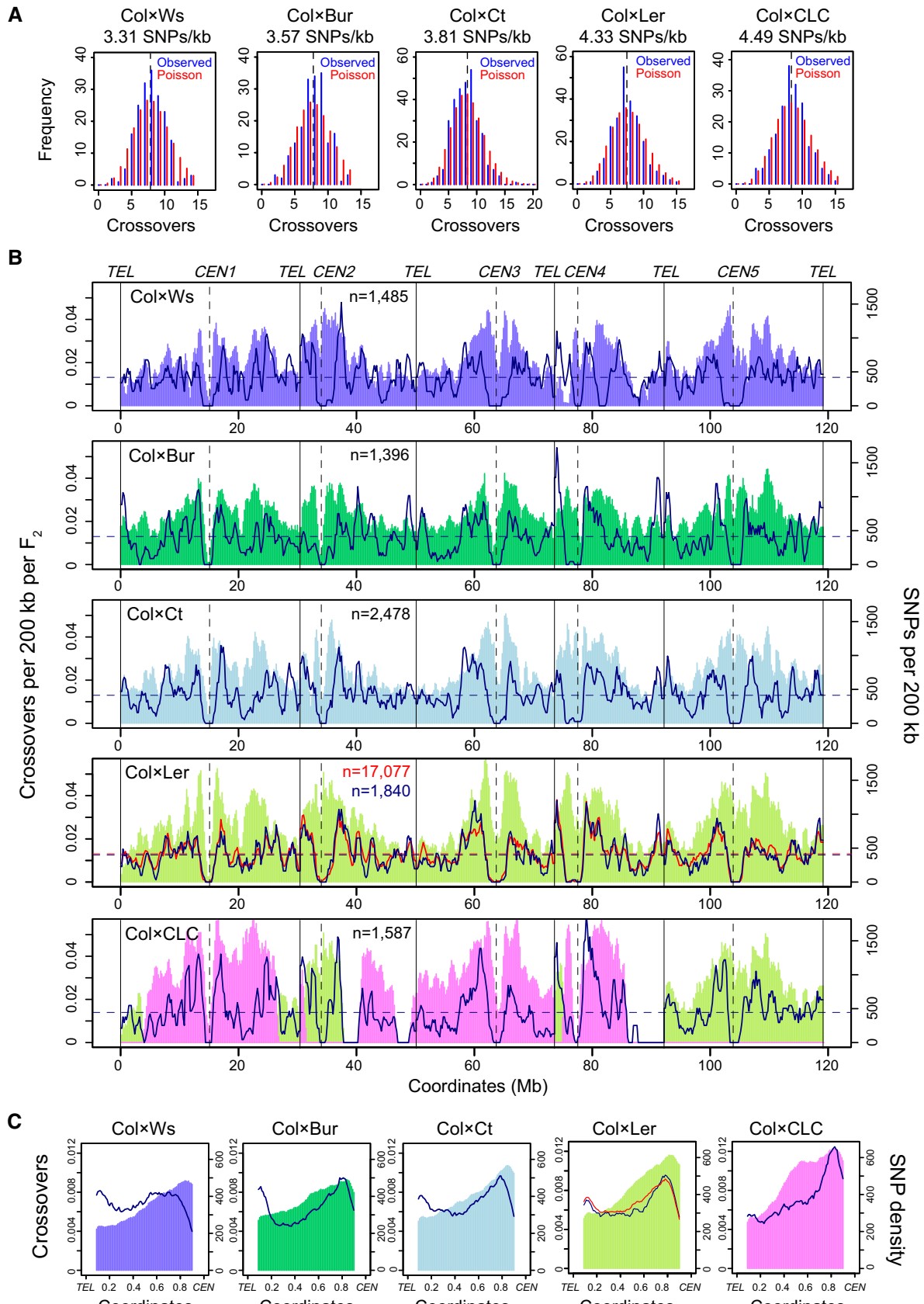

**Figure 1.**

◀

**Figure 1. Crossover and diversity landscapes in the *Arabidopsis* genome.**

A  Histograms of crossovers (blue) per $F_2$ individual in the indicated populations, with the Poisson expectation plotted in red. Mean values are indicated by the black dotted lines. The genome average SNPs/kb for each cross are printed above the plots.

B  Crossovers per 200 kb per $F_2$ plotted along the *Arabidopsis* chromosomes. Mean values are shown by horizontal dashed lines. SNPs per 200 kb are plotted and shaded in colour (Col × Ws (purple), Col × Bur (Lawrence *et al*, 2019) (dark green), Col × Ct (light blue), Col × Ler $F_2$ populations with 17,077 (red) and 1,840 (blue) crossovers (Serra *et al*, 2018b; Rowan *et al*, 2019) and Col × CLC (pink/light green, where pink indicates Cvi SNPs and green indicates Ler SNPs). The positions of telomeres (*TEL*) and centromeres (*CEN*) are indicated. The number of crossovers analysed is printed inset.

C  Data as for B, but analysing crossovers (lines) or SNPs (shading) along proportionally scaled chromosome arms, orientated from telomere (*TEL*) to centromere (*CEN*).

stable between crosses, with the mean varying between 7.51 and 8.40 per individual (Fig 1A and Appendix Table S3). The number of crossovers per $F_2$ was significantly different to the Poisson expectation in each population, as assessed by goodness-of-fit tests ($P = 0.012$–$3.63 × 10^{-6}$; Fig 1A), which is consistent with the action of crossover interference and homeostasis (Jones & Franklin, 2006). In order to investigate crossover spacing, we identified *cis*-DCOs from our $F_2$ genotyping data, by filtering for parental-heterozygous-parental genotype transitions (e.g. Col-Het-Col or Ler-Het-Ler in Col × Ler $F_2$ individuals; Appendix Fig S1) (Drouaud *et al*, 2005; Rowan *et al*, 2019; Lambing *et al*, 2020a). We generated 2,000 sets of matched randomly generated distances as a control in each case. The random distances were compared with observed DCOs using permutation tests. This showed that the distances between observed DCOs were significantly greater than the random distances, in all populations (all $P < 0.005$; Appendix Fig S1). Together, these data show stable maintenance of crossover numbers and spacing across the tested range of heterozygosity in *Arabidopsis*.

To analyse recombination and diversity landscapes throughout the Arabidopsis genome, we calculated crossover and SNP density and plotted along the chromosomes, after normalizing crossovers by the number of $F_2$ individuals sequenced (Fig 1B). The replicate Col × Ler crossover maps were significantly positively correlated (Spearman's correlation coefficient $r_s = 0.904$ at 200 kb scale; Fig 1B and Appendix Table S4), demonstrating the reproducibility of our mapping approach. Significant, yet weaker, positive correlations were observed between the other crossover maps ($r_s = 0.580$–$0.712$; Fig 1B and Appendix Table S4). Structural variation may contribute to the differences between recombination maps (Rowan *et al*, 2019). For example, Bur has a highly reduced *45S* rDNA gene copy number within *NOR2* (Rabanal *et al*, 2017), which correlates with relative suppression of Col × Bur crossover frequency in this region (Fig 1B). Additionally, a zone of crossover suppression on

the long arm of chromosome 4 in the Col × Ws map is likely to reflect the presence of a known inversion (Fig 1B) (Rowan *et al*, 2015).

We analysed scaled chromosome arms orientated from telomere to centromere and observed that crossovers showed a U-shaped distribution, to varying extents (Fig 1C). Across all maps, the highest crossover levels were observed in the sub-telomeres and pericentromeres (Fig 1C). These patterns may relate to pairing of Arabidopsis telomeres and centromeres observed during prophase I (Armstrong *et al*, 2001; Da Ines *et al*, 2012). Using the same analysis, we observed that SNPs show a progressive increase from the telomeres to the centromeres (Fig 1B and C). We observed positive correlations between crossover frequency and SNP density (e.g. Col × Ler $r_s = 0.545$; Figs 1B and 2A). Interestingly, the highly recombining pericentromeres are also the most divergent regions with the highest levels of SNPs (Fig 1B and C). The pericentromeres are defined as regions with higher than average DNA methylation, which surround the centromeres (Appendix Fig S2) (Choi *et al*, 2018). It is important to note that chromatin strongly influences meiotic recombination in Arabidopsis, which also varies along the chromosome telomere-centromere axes (Appendix Fig S2) (Yelina *et al*, 2015; Choi *et al*, 2018; Underwood *et al*, 2018; Lambing *et al*, 2020b). For example, the centromeres remained crossover suppressed in all populations (Fig 1B and C), which correlates with high levels of heterochromatic marks including DNA methylation and H3K9me2, and suppression of meiotic DSBs mapped by SPO11-1-oligo sequencing (Appendix Fig S2) (Choi *et al*, 2018; Underwood *et al*, 2018; Walker *et al*, 2018; Lambing *et al*, 2020b).

Importantly, we observed a parabolic relationship between crossover frequency and SNP density in all populations (Fig 2A). Initially, crossover frequency and SNP density show a positive correlation, but higher polymorphism density associates with

**Figure 2. A parabolic relationship between SNP density and crossover frequency.**　　　　　　　　　　　　　　　　　　　　　　　▶

A  Crossover frequency normalized by the number of $F_2$ individuals and SNP density in 100 kilobase (kb) adjacent windows were calculated for each population and ranked into percentiles according to SNP density. Centromeric regions were excluded from analysis (Underwood *et al*, 2018). The Spearman's rank correlation coefficient (*r*) between SNP density and crossover frequency is printed inset. Trend lines were fitted in ggplot using a generalized additive model (GAM) with the formula $y \sim poly(x,2)$.

B  Generalized linear model (GLM) plots showing observed Col/Ler crossovers per megabase (red) for SNP intervals grouped into hexiles or quintiles, by increasing values (left to right along the *x*-axis) of a given explanatory variable. Data were modelled with the glm2 function in R, using the binomial family with the logit link function. The formula used for the model was Crossovers ~ (SPO11-1 + nucleosomes + H3K4me3 + DNA methylation + SNPs/kb + width)$^2$. Predicted crossovers per megabase for each group of intervals were derived by sampling from the binomial distribution based on the probabilities of intervals within each group overlapping a crossover. The black box plots represent 100 samples, where the central band represents the median, the box represents the interquartile range (IQR), and the whiskers represent the minimum and maximum values that are no more than 1.5 times the IQR from the box. The final stepAIC-selected model was as follows:
Crossovers ~ SPO11-1 + nucleosomes + H3K4me3 + DNA methylation + SNPs/kb + width + SPO11-1:nucleosomes + SPO11-1:DNA methylation + SPO11-1:width + nucleosomes:H3K4me3 + nucleosomes:DNA methylation + nucleosomes:SNPs/kb + H3K4me3:DNA methylation + H3K4me3:width + DNA methylation:width + SNPs/kb:width.

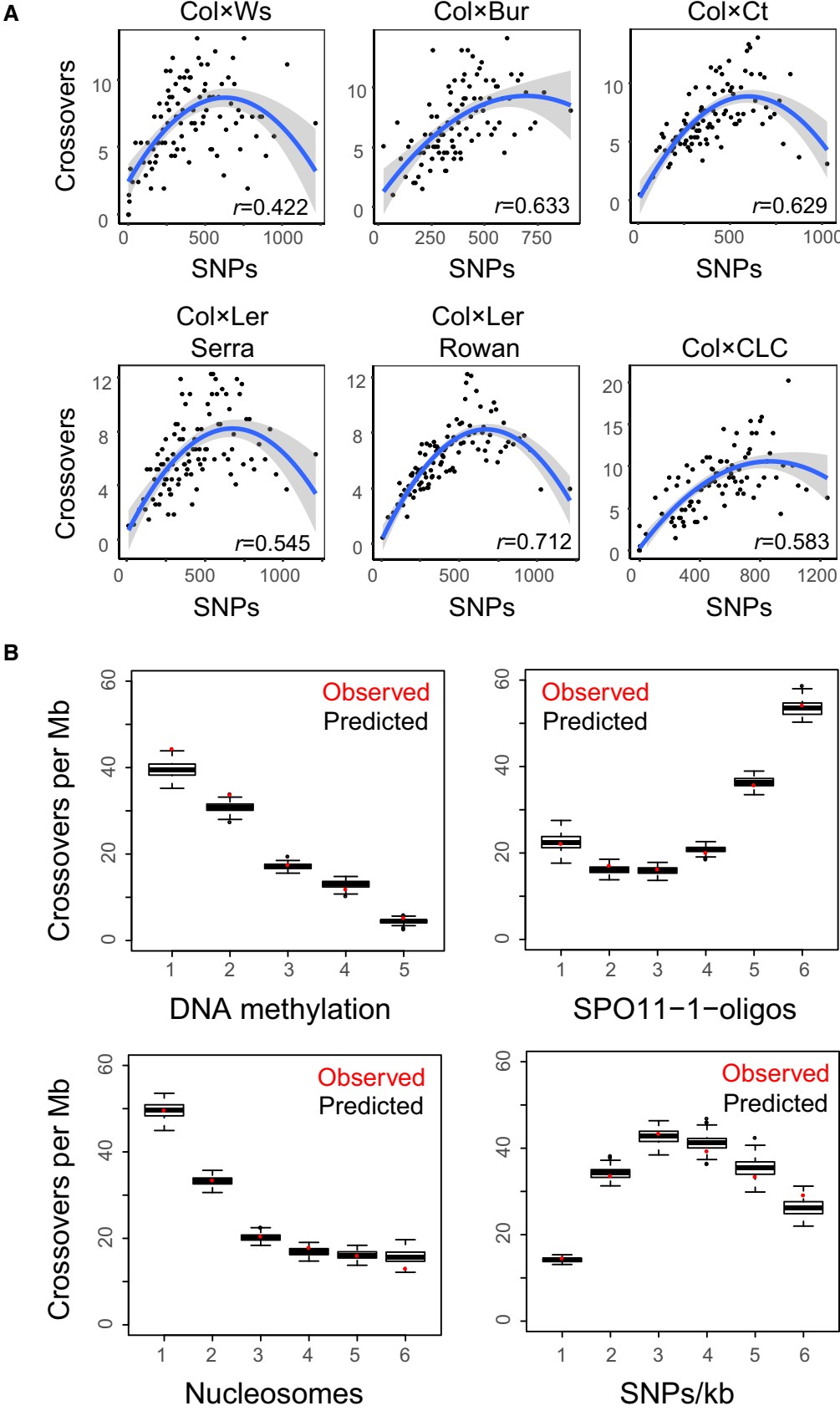

**Figure 2.**

reduced crossover frequency (Fig 2A). To quantitatively model the effects of multiple parameters on crossovers, including SNP density and chromatin features, we used a generalized linear model (GLM)

with a set of 3,320 crossovers mapped in Col/Ler F$_2$ individuals (Fig 2B and Appendix Table S5) (Choi *et al*, 2018). We considered 534,780 Col/Ler SNP intervals where it is possible to detect a

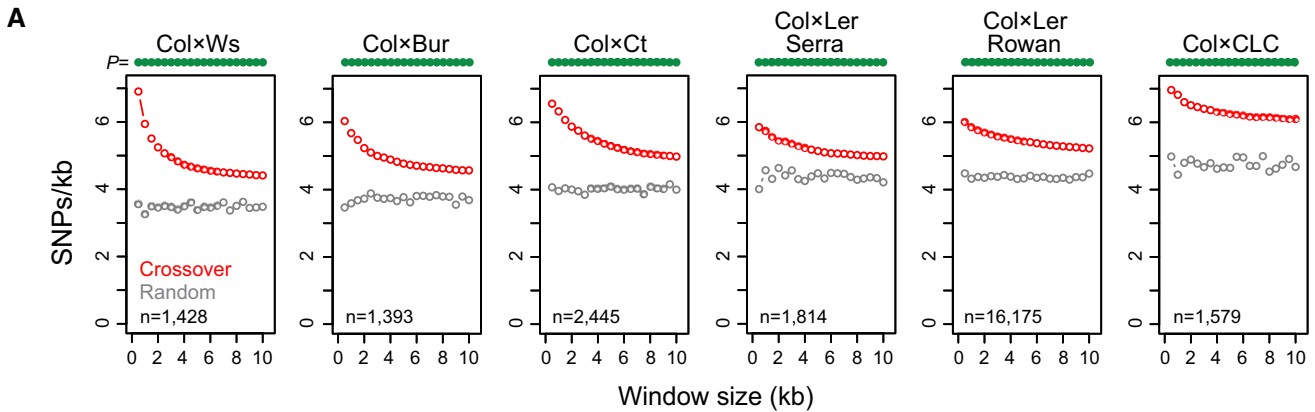

**Figure 3.**

**Figure 3. Crossovers are positively associated with SNP density at the local scale.**

A SNPs/kb in physical windows of increasing size (kb) around crossover midpoints (red), or for matched randomly chosen positions (grey). Printed above the plot for each window are circles coloured green if crossover SNPs/kb values are significantly different to random ($P < 0.05$), or red if not ($P > 0.05$; Bonferroni-adjusted $t$-tests). The population (Col × Ws, Col × Bur (Lawrence *et al*, 2019), Col × Ct, Col × Ler (Serra *et al*, 2018b; Rowan *et al*, 2019) and Col × CLC) is printed above each plot, and the number of positions analysed is printed inset.

B As for A, but analysing the indicated Col/Ler transition and transversion polymorphisms around Col × Ler crossovers (Rowan *et al*, 2019) (red), or the same number of random positions (grey).

C As for A, but analysing Col/Ler SNPs/kb around crossovers from wild type, *HEI10-OE*, *recq4a recq4b* or *HEI10-OE recq4a recq4b* (Serra *et al*, 2018b). All populations were generated from Col × Ler hybrids. Printed beneath are the estimated average numbers of Class I and Class II crossovers per meiosis for each genotype, measured via genotyping-by-sequencing (Ziolkowski *et al*, 2017; Serra *et al*, 2018b). To the right are plots showing the significance of SNPs/kb differences between crossover sets, across the windows tested (Bonferroni-adjusted $t$-tests).

crossover, which had a mean width of 224 bp. The binary response variable in the model is whether at least one crossover was observed in a given SNP interval. We then calculated explanatory variables for the same intervals, including SPO11-1-oligos, nucleosomes (MNase-seq), H3K4me3 (ChIP-seq), DNA methylation (BS-seq) and SNP density (Choi *et al*, 2018). For SNP density, we calculated a rolling average of SNPs/kb with a one base pair step and used these values to calculate mean SNPs/kb per interval. Data were modelled using the binomial family with a logit link function and the formula, Crossovers ~ (SPO11-1-oligos + nucleosomes + H3K4me3 + DNA methylation + SNPs/kb + width)$^2$.

The formula for the final model was selected based on lowest Akaike information criterion (AIC). The model shows a negative effect of higher nucleosomes and DNA methylation on crossovers, and a positive effect for higher SPO11-1-oligos (Fig 2B and Appendix Table S5) (Choi *et al*, 2018). We again observed that SNP density shows a parabolic relationship with crossovers (Fig 2B). Initially, a positive relationship is observed with increasing SNPs/kb associating with higher crossover frequency (Fig 2B). However, beyond a certain polymorphism threshold the relationship becomes negative (Fig 2B). Together, this is consistent with high levels of SNPs and structural polymorphism causing crossover suppression in Arabidopsis (Serra *et al*, 2018a; Rowan *et al*, 2019). In summary, across these maps we see evidence that, below a critical threshold, regions of elevated SNP density attract crossovers, in addition to a strong effect of chromatin and meiotic DSB frequency.

## Crossovers associate with higher SNP density at the kilobase scale

Due to the pericentromeric regions showing both elevated SNP density and crossover frequency (Fig 1B and C), we sought to examine polymorphism density in relation to crossover sites at the local (kilobase) scale. We analysed SNP density (SNPs/kb) in windows of increasing physical size around crossover midpoints, divided by the number of crossovers analysed in each data set (Fig 3A). Analysis was restricted to crossovers resolved between SNPs less than 10 kb apart. As a control, a matched number of randomly selected positions were analysed (Fig 3A). In each population, we observed significant enrichment of SNPs/kb values around the crossovers compared with random, for each window size tested (Bonferroni-adjusted $t$-tests, all $P < 5.04 \times 10^{-5}$; Fig 3A). We repeated this analysis after separating crossovers into those located within the pericentromeres versus the chromosome arms (Appendix Fig S3). We

observed similar trends to the genome-wide analysis, with crossovers from both regions associating with significantly higher SNPs/kb compared with random in the majority of window sizes (Appendix Fig S3). We further analysed Col/Ler transitions and transversions separately, using the larger Col × Ler crossover data set ($n = 16,175$; Fig 3B). We observed that each polymorphism class showed significant enrichment around crossovers compared with random positions, over all windows tested (Bonferroni-adjusted $t$-tests, all $P < 1.85 \times 10^{-4}$; Fig 3B). Together, these analyses indicate that crossovers are positively associated with interhomolog polymorphism in *Arabidopsis* at the kilobase scale. Finally, we analysed base composition around crossovers and observed AT enrichment in all populations, compared with random positions (Appendix Fig S4). As reported, AT sequence enrichment at crossovers correlates with locally reduced nucleosome occupancy and elevated levels of SPO11-1-oligos (Appendix Fig S4) (Choi *et al*, 2018).

## The Class I repair pathway mediates crossover association with elevated SNP density

We sought to investigate the genetic requirements of crossover association with higher sequence polymorphism. Previously, we generated crossover maps using populations where the Class I repair pathway was increased via overexpression of the HEI10 E3 ligase (*HEI10-OE*), or the Class II pathway was increased via *recq4a recq4b* mutations, or Class I and Class II pathways were simultaneously increased in *HEI10-OE recq4a recq4b* (Séguéla-Arnaud *et al*, 2015; Ziolkowski *et al*, 2017; Serra *et al*, 2018b). All populations were generated from Col × Ler F$_1$ hybrids (Ziolkowski *et al*, 2017; Serra *et al*, 2018b). We analysed SNPs/kb enrichment around crossovers from wild type, *HEI10-OE*, *recq4a recq4b* or *HEI10-OE recq4a recq4b* and compared with the same number of random positions in each case (Fig 3C). In both wild type and *HEI10-OE*, where Class I repair is increased, crossovers show significant enrichment of SNPs/kb compared with random, over the majority of window sizes (Bonferroni-adjusted $t$-tests, $P < 1.40 \times 10^{-4}$; Fig 3C). SNPs/kb enrichment around wild type and *HEI10-OE* were not significantly different from one another, for all windows up to 8 kb (Fig 3C). In contrast, in *recq4a recq4b* and *HEI10-OE recq4a recq4b*, where Class II repair is increased, we observed that SNPs/kb around crossovers were significantly reduced compared with wild type, across all windows tested (Bonferroni-adjusted $t$-test all $P < 1.87 \times 10^{-7}$; Fig 3C). Together, these analyses are consistent with crossovers associating with regions of higher SNP density via the Class I pathway.

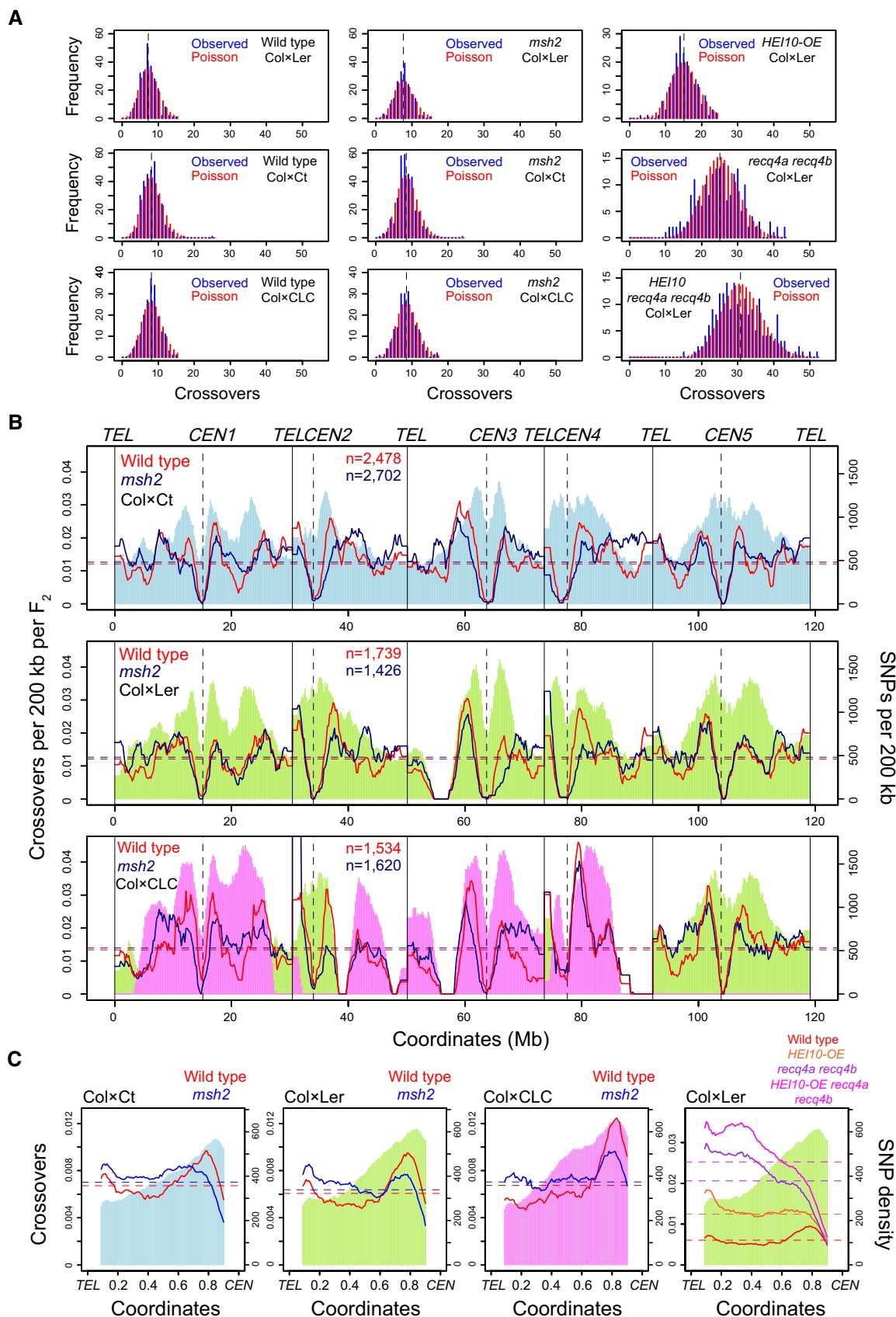

Figure 4.

**Figure 4. Remodelling of the crossover landscape in *msh2*.**

A  Histograms of observed crossovers per $F_2$ individual in the indicated wild type and *msh2* populations, from Col × Ct, Col × Ler and Col × CLC crosses (blue). The Poisson expectation is plotted in red. Data from *HEI10-OE*, *recq4a recq4b* and *HEI10-OE recq4a recq4b* $F_2$ populations are also shown for comparison (Ziolkowski *et al*, 2017; Serra *et al*, 2018b). Mean values are indicated by black dotted lines.

B  Crossovers per 200 kb per $F_2$ plotted along the Arabidopsis chromosomes, with mean values shown by horizontal dashed lines. Data are shown for wild type (red) and *msh2* (blue) crossovers generated from Col × Ct, Col × Ler and Col × CLC hybrids. SNPs per 200 kb are shaded in colour (blue = Col/Ct, green = Col/Ler, pink = Col/Cvi). The positions of telomeres (*TEL*) and centromeres (*CEN*) are labelled. The number of crossovers analysed is printed inset.

C  Data as for B, but analysing crossovers in wild type (red) and *msh2* (blue), or SNPs along proportionally scaled chromosome arms orientated from telomeres (*TEL*) to centromeres (*CEN*). Crossover data from *HEI10-OE*, *recq4a recq4b* and *HEI10-OE recq4a recq4b* Col × Ler $F_2$ populations are also shown (Ziolkowski *et al*, 2017; Serra *et al*, 2018b).

## The crossover landscape remodels towards lower diversity regions in *msh2*

We next tested whether interactions between polymorphism and crossover frequency are dependent on the mismatch sensor MSH2 in Arabidopsis. We first backcrossed the *msh2-1* T-DNA insertion (Leonard *et al*, 2003) from the Col accession into the Ler and CLC backgrounds for five generations, maintaining *msh2-1* as a heterozygote during this process, in order to minimize mutation accumulation. We also used CRISPR/Cas9 mutagenesis to generate a *msh2* allele *de novo* in the Ct accession (Appendix Fig S5). A pair of gRNAs targeting *MSH2* exon four were designed and introduced upstream of the *U3* and *U6* promoters. These constructs were transformed into Ct, together with an *ICU2::Cas9* transgene. Transformed $T_1$ plants were genotyped by PCR amplification with primers flanking the *MSH2* gRNA target sites, and sequencing was performed to detect deletions. A *msh2* mutant with a heritable frame shift deletion that did not carry the CRISPR-Cas9 transgenes was selected for further experiments (*msh2-3*; Appendix Fig S5). The *msh2* mutants in the Ler, Ct and CLC backgrounds were then crossed to the *msh2-1* Col line to generate $F_1$ hybrid plants that were *msh2* homozygous mutants, and these hybrids were then self-fertilized to generate $F_2$ progeny. We sequenced 187, 320 and 191 *msh2* $F_2$ individuals in the Col × Ler, Col × Ct and Col × CLC backgrounds and mapped 1,426, 2,702 and 1,620 crossovers, respectively (Fig 4A and Appendix Table S3).

A slight but not significant increase in crossover numbers per $F_2$ individual was observed in each *msh2* population, compared with wild type (*t*-test Col × Ler $P = 0.08$, Col × Ct $P = 0.12$ and Col × CLC $P = 0.16$; Fig 4A). This is in contrast to the large crossover increases per $F_2$ observed in *HEI10-OE*, *recq4a recq4b* and *HEI10-OE recq4a recq4b* populations (Fig 4A) (Ziolkowski *et al*, 2017; Serra *et al*, 2018b). However, we observed significant changes to the *msh2* crossover landscape at the chromosome scale. Specifically, *msh2* crossovers were depleted in the pericentromeres and increased in the chromosome arms (chi-squared test Col × Ler $P = 1.69 \times 10^{-7}$, Col × Ct $P = 2.76 \times 10^{-15}$ and Col × CLC $P = 1.00 \times 10^{-11}$; Fig 4B and C and Appendix Table S6). We tested scaled windows along the proportional length of all chromosome arms, from telomeres (*TEL*) to centromeres (*CEN*), and used a Poisson model to compare crossover counts between wild type and *msh2* (Appendix Fig S6). This confirmed that pericentromeric windows had significantly decreased crossovers in *msh2*, whereas sub-telomeric and interstitial windows had increased crossovers ($-\log_{10}$ (BH-adjusted *P*-values) > 1; Appendix Fig S6).

We calculated crossover and SNP density in 100 kilobase (kb) adjacent windows, normalized crossovers by the number of $F_2$ individuals and grouped into percentiles. As noted previously, a positive correlation between crossovers and SNPs was observed using these percentiles in the wild-type populations (Col × Ler $r_s = 0.545$, Col × Ct $r_s = 0.629$, Col × CLC $r_s = 0.583$), which was absent or reduced in *msh2* (Col × Ler $r_s = $ n.s., Col × Ct $r_s = $ n.s, Col × CLC $r_s = 0.250$; Appendix Fig S7A). In comparison the correlation between crossovers and SNPs is strongly negative in *recq4a recq4b* ($r_s = -0.772$) and *HEI10 recq4a recq4b* ($r_s = -0.784$), where Class II repair is increased (Appendix Fig S7B) (Fernandes *et al*, 2017; Serra *et al*, 2018b). Interestingly, the correlation between polymorphism and crossovers is not significant in *HEI10-OE* (Appendix Fig S7B). This is likely as a consequence of increased Class I activity, for example in *HEI10-OE* and during male meiosis, associating with elevated distal crossovers in regions of lower SNP density (Fig 4C) (Fernandes *et al*, 2017; Ziolkowski *et al*, 2017). Hence, *msh2* causes remodelling of the crossover landscape and an altered relationship with interhomolog diversity.

To confirm changes to the *msh2* crossover landscape, we utilized fluorescent tagged lines (FTLs; Appendix Fig S8 and Tables S7–S9). FTL intervals are defined by T-DNA insertions that express different colours of fluorescent protein (green or red) in pollen (*LAT52* promoter) or seed (*NapA* promoter) (Francis *et al*, 2007; Wu *et al*, 2015). When linked T-DNAs are hemizygous, patterns of fluorescence in pollen or seed can be used to quantify crossover frequency within the interval flanked by the T-DNAs (Francis *et al*, 2007; Wu *et al*, 2015). We used the *I1b* FTL that measures crossover frequency within an interstitial region on chromosome 1 (*I1b*), located 3.9 Mb from the telomere. *I1b* showed a significant crossover increase in *msh2* hybrids compared with wild type (*t*-tests Col/Ler $P = 4.37 \times 10^{-6}$ and Col/CLC $P = 1.34 \times 10^{-8}$; Appendix Fig S8 and Table S7). To compare centromeric regions, we tested intervals that span the centromere on chromosomes 5 (*5.10*) and 3 (*CEN3*; Appendix Fig S8 and Tables S8 and S9). We observed that both *CEN3* and *5.10* showed significant crossover decreases in *msh2* hybrids compared with wild type (*t*-tests *5.10* Col/Ler $P = 0.017$ and Col/CLC $P = 0.032$; *CEN3* Col/Ct $P = 3.31 \times 10^{-6}$; Appendix Fig S8 and Tables S8 and S9). These trends are consistent with our sequencing-based crossover maps, where crossovers remodelled from the pericentromeres towards the distal regions in *msh2* hybrids. Interestingly, *I1b* in Col/CLC hybrids showed a greater relative increase in *msh2* when measured via FTL compared with the sequencing-based estimates. This may be due to *I1b* measuring male crossover frequency, where recombination is elevated within the sub-telomeres (Giraut *et al*, 2011), whereas the GBS-derived crossover maps represent both male and female meiosis. Therefore, the Col/CLC background may be particularly sensitive to mutations causing crossover distalization.

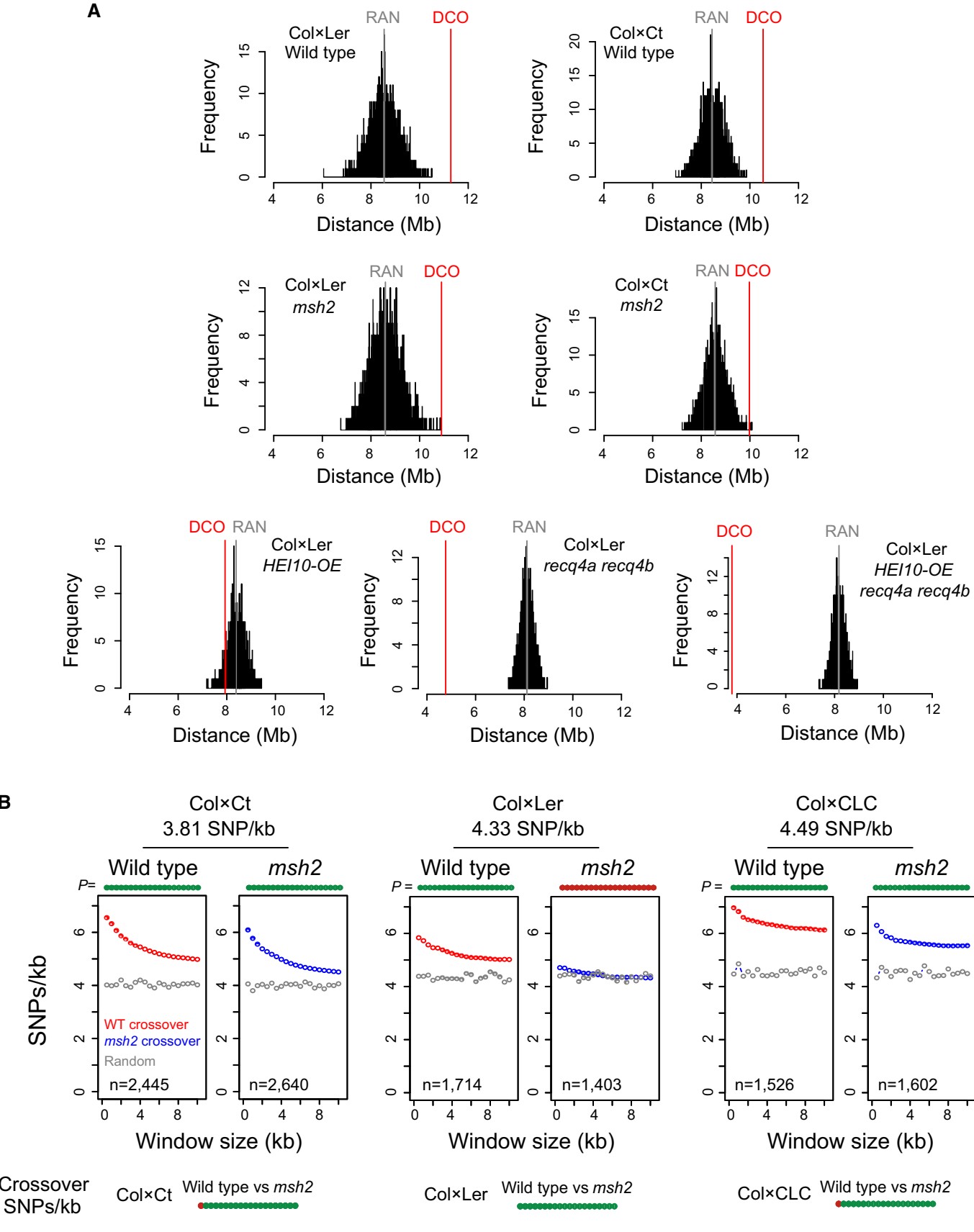

Figure 5.

**Figure 5.  Crossover association with polymorphism is reduced in *msh2* at the local scale.**

A   Histograms showing the mean distance (megabases, Mb) of observed double crossovers (DCO, red vertical line), compared with 2,000 matched sets of randomly generated distances (RAN, black). The mean distance of the random sets is shown in by the vertical grey line.

B   SNPs/kb values were calculated in physical windows of increasing size (kb) around crossover midpoints in wild type (red) and *msh2* (blue). The same values were calculated for matched randomly chosen positions (grey). Printed above the plot for each window are circles that are coloured green if crossover SNPs/kb values are significantly different to random ($P < 0.05$), or red if not ($P > 0.05$; Bonferroni-adjusted $t$-tests). The population analysed (Col × Ct, Col × Ler, Col × CLC) and its genome average SNP frequency (SNPs/kb) is printed above the plots and the number of crossover positions analysed is printed inset. Beneath are plots showing the significance of SNPs/kb differences around crossovers between the indicated genotypes.

## Local crossover association with SNPs is altered in *msh2*

As described previously, we identified *cis*-DCOs from our $F_2$ genotyping data, by filtering for parental-heterozygous-parental genotype transitions (Drouaud *et al*, 2005; Rowan *et al*, 2019; Lambing *et al*, 2020a), and recorded DCO distances (Fig 5A and Appendix Fig S9). Two thousand sets of matched random distances were generated for each population, which were compared with observed DCO distances (Fig 5A and Appendix Fig S9). The CLC populations were not analysed in this way, due to the introgression structure of this background complicating DCO identification. DCOs in both the wild type and *msh2* Col × Ler and Col × Ct populations were significantly greater than random (permutation tests, both $P < 0.005$), consistent with normal crossover interference. In both Col × Ler and Col × Ct populations, *msh2* DCO distances were slightly but not significantly reduced relative to wild type, indicating that interference is normal in *msh2* (Fig 5A). By comparison, greater reductions in crossover spacing were observed in *HEI10-OE*, *recq4a recq4b* and *HEI10-OE recq4a recq4b* (Fig 5A).

We analysed SNP density at the local scale around crossovers in wild type and *msh2* and compared with the same number of random positions (Fig 5B). Relative to wild type, we observed that *msh2* showed significantly reduced SNP enrichment around crossovers, in all windows greater than 1 kb in the Col × Ler, Col × Ct and Col × CLC backgrounds (Bonferroni-adjusted $t$-tests, all $P < 0.023$; Fig 5B). The reduction in SNP association around crossovers in *msh2* occurred to varying degrees in the Col × Ler, Col × Ct and Col × CLC populations, which did not correlate with the polymorphism level in each cross (Fig 5B). We compared wild-type and *msh2* crossovers with respect to base frequency, nucleosome occupancy and SPO11-1-oligos (Appendix Fig S10). We observed that crossovers in *msh2* showed AT sequence enrichment, reduced nucleosome occupancy and elevated SPO11-1-oligos, which was similar to wild-type crossovers (Appendix Fig S10) (Choi *et al*, 2018). Together, this is consistent with *msh2* crossovers forming in regions of elevated meiotic DSBs, but with a reduced association with SNP density relative to wild type.

To test whether *msh2* influences crossover formation in relation to large structural polymorphisms, we measured how frequently crossovers overlapped a set of 47 high confidence Col/Ler inversions (total length = 1.59 Mb, mean inversion width = 33.8 kb) (Zapata *et al*, 2016). In wild-type Col × Ler, 2 of 1,739 crossovers overlapped an inversion, and no overlaps were observed with 1,426 *msh2* crossovers. Using matched random windows with the same widths as the crossovers, 25 and 19 overlaps were observed with the inversions, which were significantly greater than the observed crossover overlaps (chi-squared test wild type $P = 2.13 \times 10^{-5}$ and *msh2* $P = 3.42 \times 10^{-5}$). Hence, *msh2* does not significantly increase crossovers within large inversions in *Arabidopsis*.

## MSH2 associates with meiotic chromatin during early prophase I

To screen for meiotic phenotypes in *msh2* hybrids, we performed chromosome spreads and stained chromatin with DAPI (Fig 6A). The *msh2* mutant showed no detectable phenotypes compared with wild type, during pachytene, diakinesis, metaphase I, dyad and tetrad stages, in either the Col × Ler or Col × CLC hybrid backgrounds (Fig 6A). The proportion of rod and ring bivalents at metaphase I can be used to estimate one versus greater than one crossovers per chromosome. Consistent with our $F_2$ sequencing data (Fig 4A), wild type and *msh2* did not show significant differences in ring and rod counts in Col × Ler or Col × CLC hybrids (Wilcoxon rank-sum tests, $P = 0.967$ and $P = 0.234$; Appendix Table S10). To assess fertility, we performed Alexander staining of pollen and observed small but significant decreases in *msh2* pollen viability (Appendix Table S11), although high levels of fertility were observed for both inbred and hybrid *msh2* (> 90% viability). Hence, remodelling of crossovers with respect to polymorphism in *msh2* does not cause sterility or cytologically detectable meiotic defects in *Arabidopsis*. Therefore, the decreased fertility of inbred *msh2* mutants is likely due to mutations accumulating during somatic development (Leonard *et al*, 2003; Hoffman *et al*, 2004; Watson *et al*, 2016; Belfield *et al*, 2018).

In order to test whether MSH2 associates with meiotic chromatin, we performed immunocytology in wild type and *msh2*. We spread *Arabidopsis* male meiocytes and immunostained for MSH2 and the meiotic axis HORMA domain protein ASY1 (Armstrong *et al*, 2002) and stained chromatin with DAPI (Fig 6B). Nuclei in early prophase I were identified by linear ASY1 signal, which is coincident with meiotic DSB formation (Armstrong *et al*, 2002; Sanchez-Moran *et al*, 2007). These nuclei showed that MSH2 signal tracked the axes and showed punctate higher abundance foci, which were not detectable in *msh2* (Fig 6B). This is consistent with a role for MSH2 in regulating DSB repair during early prophase I. MSH4 is related to MSH2 and plays a key role in the Class I pathway to promote interfering crossovers (Higgins *et al*, 2004). Therefore, we co-immunostained for MSH2 and MSH4 during early prophase I (Fig 6C). This revealed that both MSH2 and MSH4 associate with meiotic chromatin as punctate foci during early prophase I (Fig 6C). We observed a mean of 186 MSH2 foci per cell, of which 131 (74%) overlapped MSH4 foci. As a control for co-localization, the MSH2 images were rotated 180 degrees and the number of foci overlapping MSH4 foci was quantified. Following rotation, significantly fewer MSH2 foci overlapped MSH4 foci (mean = 65 foci, 36%; $t$-test $P = 2.25 \times 10^{-3}$), which supports that MSH2 and MSH4 significantly co-localize on *Arabidopsis* meiotic chromosomes. These data are consistent with MSH2 associating with meiotic chromatin and regulating crossover repair of interhomolog recombination intermediates.

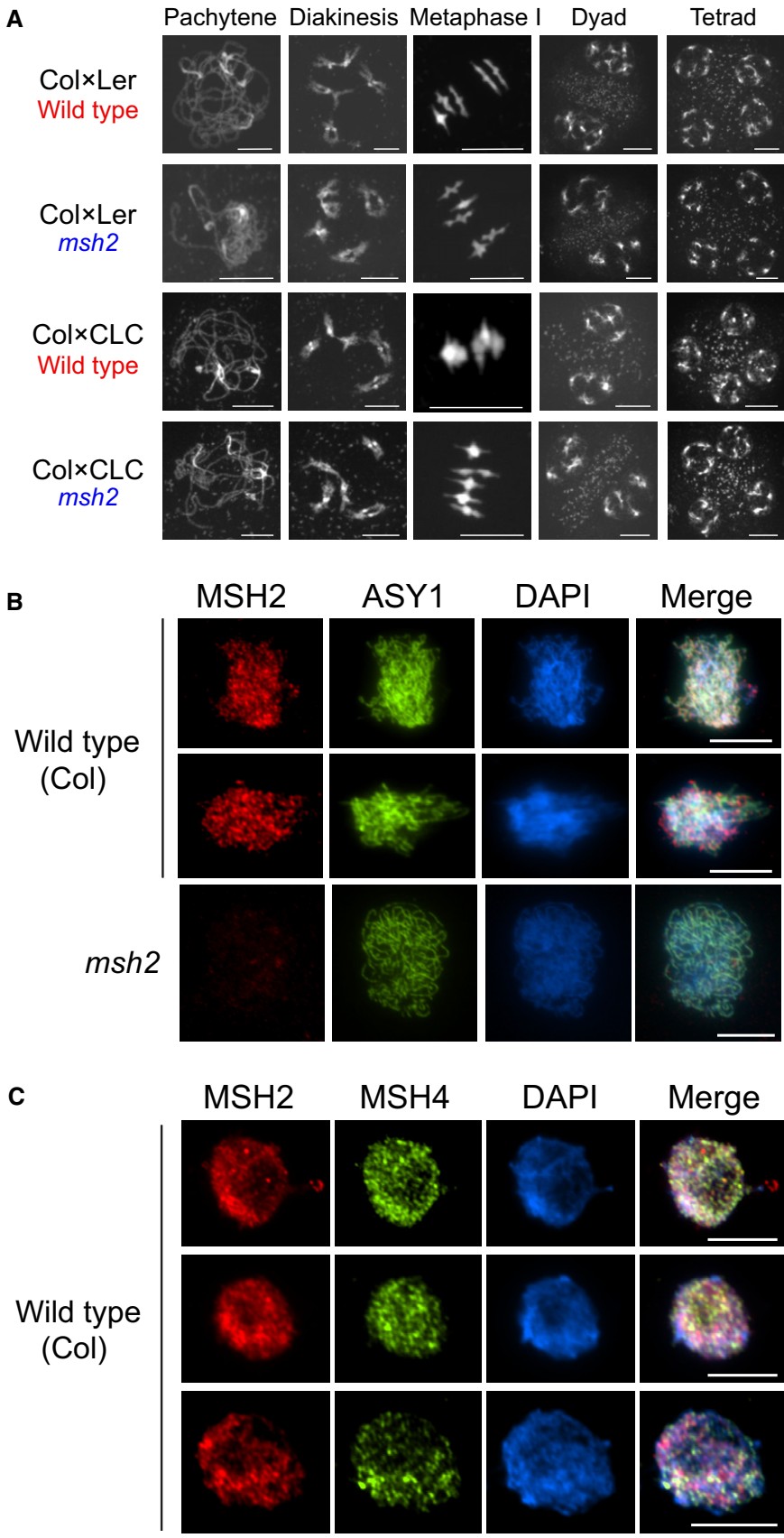

**Figure 6.**

**Figure 6. MSH2 accumulates on meiotic chromatin during early prophase I.**

A Representative DAPI-stained spreads of pachytene, diakinesis, metaphase I, dyad and tetrad meiotic stages, for wild type and *msh2*, in Col × Ler or Col × CLC hybrid backgrounds.

B Male meiocytes immunostained for MSH2 (red) and ASY1 (green), and stained for DAPI (blue) in wild type (Col) or *msh2*.

C As for B, but immunostaining wild type male meiocytes for MSH2 (red), MSH4 (green) and staining chromatin with DAPI (blue).

Data information: All scale bars indicate 10 μm.

## Crossover remodelling into heterozygous regions is *MSH2* dependent

In Arabidopsis, juxtaposition of heterozygous and homozygous regions, at the megabase scale, causes crossover increases in the heterozygous region, at the expense of the adjacent homozygous region (Ziolkowski *et al*, 2015). This juxtaposition effect can be detected by combining FTL crossover reporters with recombinant lines (Fig 7A and B) (Ziolkowski *et al*, 2015). For example, the *420* FTL interval measures crossover frequency in a 5 megabase sub-telomeric region on chromosome 3, which was previously shown to respond to the juxtaposition effect (Fig 7A and B) (Ziolkowski *et al*, 2015). *420* crossover frequency was measured in one of four Col/Ct polymorphism configurations: (i) "HOM-HOM" that are Col/Col inbred throughout the genome, (ii) "HET-HET" that are Col/Ct heterozygous throughout the genome, (iii) "HET-HOM" where the *420* region is Col/Ct heterozygous and the remainder of chromosome 3 is Col/Col homozygous and (iv) "HOM-HET" where *420* is Col/Col homozygous and the remainder of chromosome 3 is Col/Ct heterozygous (Fig 7B and Appendix Fig S11). Col × Ct hybrids were chosen for these experiments as there is an absence of *trans*-acting recombination modifier loci in this cross (Ziolkowski *et al*, 2015). As reported, HET-HOM lines show a significant increase in *420* crossover frequency compared with HOM-HOM (*t*-test $P = 5.57 \times 10^{-15}$), whereas HOM-HET show a significant decrease (*t*-test $P = < 2.2 \times 10^{-16}$; Fig 7C and Appendix Table S12). This is consistent with heterozygosity promoting crossover recombination when juxtaposed with homozygosity (Ziolkowski *et al*, 2015).

The heterozygosity juxtaposition effect represents a context where sequence divergence promotes crossover formation. We therefore sought to test whether this phenomenon is *MSH2* dependent. For this purpose, we employed our Col/Ct recombinant lines (Fig 7A and B) and used CRISPR-Cas9 to introduce null mutations in *MSH2* (Appendix Fig S11). As described earlier, Arabidopsis Col and Ct parental and recombinant lines were transformed with constructs expressing Cas9 together with pairs of CRISPR gRNAs targeting exon 4 of *MSH2*, to generate independent *msh2* mutations (*msh2-2*, *msh2-3*, *msh2-4* and *msh2-5*; Appendix Fig S5). All four *msh2* alleles result from deletions that cause premature stop codons (Appendix Fig S5). We repeated crossover frequency measurements in these lines and observed that *msh2* caused opposite trends in recombination to those seen in wild type. Specifically, the HET-HOM *msh2* lines showed lower *420* crossover frequency compared with HOM-HOM *msh2* (*t*-test $P = 1.92 \times 10^{-3}$), and HOM-HET *msh2* showed higher crossovers than HOM-HOM *msh2* (*t*-test $P = 2.65 \times 10^{-6}$; Fig 7C and Appendix Table S12). It is also notable that HET-HET lines in wild type showed significantly decreased *420* crossover frequency compared with HOM-HOM (*t*-test $P = 1.83 \times 10^{-7}$), but no change is evident for the same comparison in *msh2* (*t*-test $P = 0.231$; Fig 7C and Appendix Table S12).

Together, this shows that MSH2 is required to promote crossovers in heterozygous regions when they are juxtaposed with homozygous regions in Arabidopsis, providing further evidence for a pro-crossover role for MSH2 in regions of higher divergence.

We repeated the analysis of *420* heterozygosity juxtaposition in the *HEI10-OE* background, where Class I crossover repair is increased (Fig 7D, Appendix Fig S12 and Table S13) (Ziolkowski *et al*, 2017; Serra *et al*, 2018b). In this case, a Col *HEI10-OE* transgenic line was crossed with the Col and Ct recombinant lines (Appendix Fig S12). We observed that *HEI10-OE* HET-HOM and HOM-HET lines showed significant increases and decreases of crossovers respectively, compared with *HEI10-OE* HOM-HOM (*t*-test $P = 1.66 \times 10^{-15}$ and $P = 7.26 \times 10^{-12}$; Fig 7D and Appendix Table S13). These are the same trends as observed in wild type, albeit with overall elevated levels of crossover frequency (Fig 7D and Appendix Table S13). This is further consistent with crossover association with heterozygous regions being promoted via the Class I repair pathway.

## Discussion

Our results reveal an unexpected pro-crossover role for MSH2 in the Arabidopsis pericentromeric regions, which show relatively high sequence divergence. We show that crossover remodelling occurring due to the juxtaposition of heterozygous and homozygous regions is also MSH2 dependent. To explain these effects, we propose two models where MSH2 heterodimers bind mismatches in D-loop structures that occur following meiotic interhomolog strand invasion (Fig 8A and B), which is consistent with the known biochemical activity of human MSH2-MSH6 heterodimers (Honda *et al*, 2014). We note that as Arabidopsis MSH2 forms heterodimers with MSH3, MSH6 and MSH7 *in vitro* (Culligan & Hays, 2000; Adé *et al*, 2001; Wu *et al*, 2003), it is possible that MSH2 sub-complexes mediate recognition of different polymorphism classes during meiotic recombination. In the first model, we propose that Arabidopsis MSH2 may directly or indirectly recruit components of the Class I pathway to mismatched interhomolog strand invasion events and increase the chance of crossover repair (Fig 8A). Abundant evidence connects MSH2 complexes with recruitment of MutLα (MLH1/PMS1) heterodimers, for example during post-replicative mismatch correction (Modrich & Lahue, 1996; Kunkel & Erie, 2005). Notably, the MutLγ (MLH1/MLH3) heterodimer is also a component of the Class I crossover repair pathway (Mercier *et al*, 2015; Pyatnit-skaya *et al*, 2019). Therefore, activated MSH2 heterodimers may recruit MLH1/MLH3 and promote Class I crossover repair of strand invasion events in heterozygous regions in Arabidopsis (Fig 8A). Consistent with this model, budding yeast Msh2/Msh3 heterodimers have been shown to stimulate the nuclease activity of Mlh1/Mlh3 *in vitro* (Rogacheva *et al*, 2014). Alternatively, MSH2 binding may

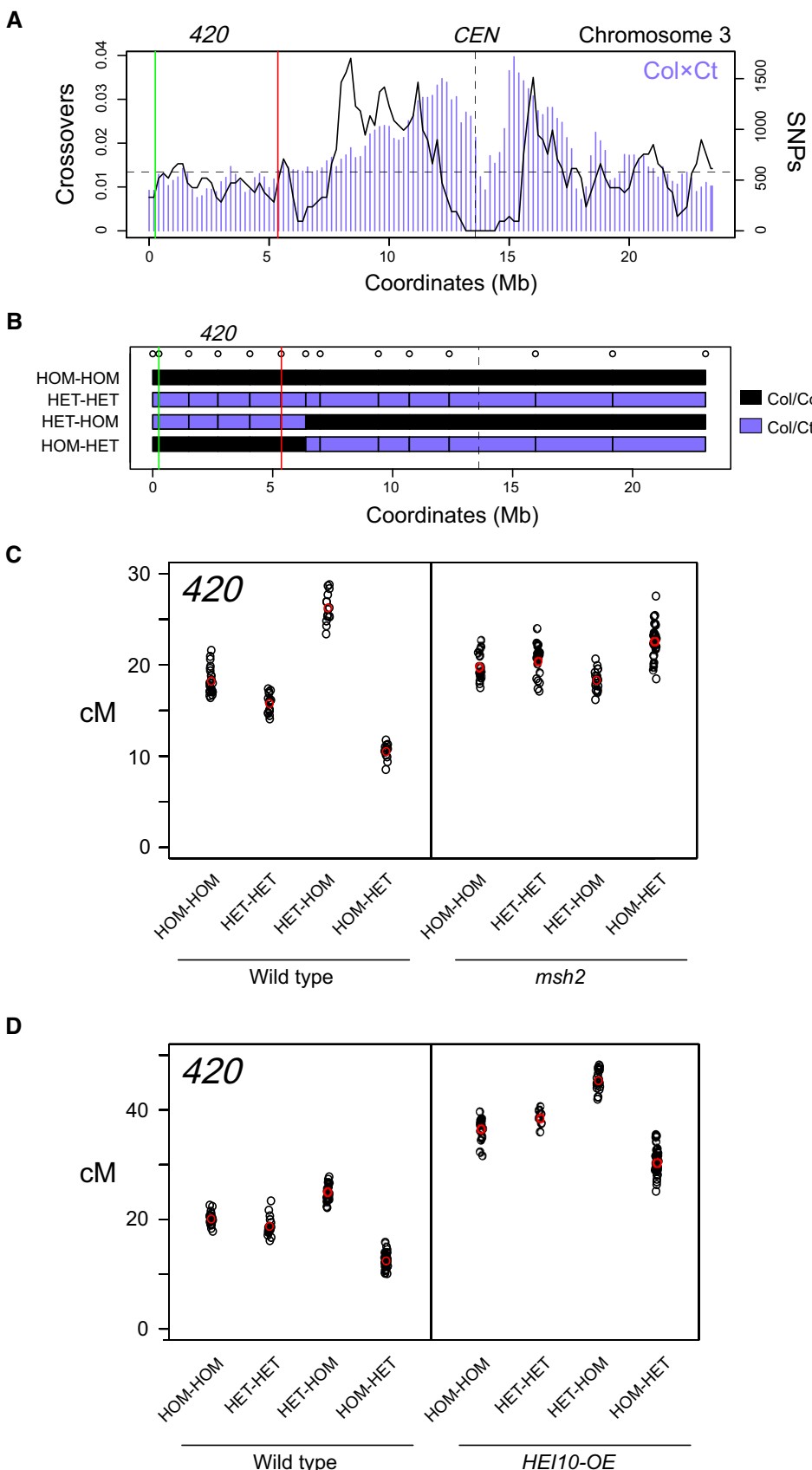

Figure 7.

**Figure 7. Crossover remodelling via juxtaposition of heterozygous and homozygous regions is *MSH2* dependent.**

A  Crossover frequency (black) and SNP density (purple) per 200 kb measured from the Col × Ct $F_2$ population. The position of the centromere (*CEN*, vertical dashed line) and *420* FTL transgenes (red and green vertical lines) used to measure crossover frequency is indicated.

B  Plots of chromosome 3 showing positions of Col/Col (black) or Col/Ct (purple) homozygous or heterozygous genotypes at the indicated marker positions (circles). The positions of the *420* FTL transgenes are indicated (red and green vertical lines).

C  *420* crossover frequency (cM) in the HOM-HOM, HET-HET, HET-HOM, HOM-HET genotypes shown in B, in either wild type or *msh2*. Values for replicate individuals are plotted, in addition to the mean (red).

D  As for C, but comparing genotypes in wild-type or *HEI10-OE* backgrounds. Values for replicate individuals are plotted, in addition to the mean (red).

promote rejection or slowed repair of mismatched strand invasion events. For example, budding yeast Msh2 recruits the Sgs1 helicase to promote the disassembly of mismatched D-loops (Myung *et al*, 2001; Mazina *et al*, 2004). Inhibited recombination has the potential to cause feedback signalling to SPO11-1 complexes and increase local meiotic DSB formation (Fig 8B). In the second model, the resulting increase in DSBs would then cause higher crossovers in regions of relatively high sequence divergence (Fig 8B). Consistent with this model, budding yeast *zip1*, *zip3* and *msh5* (*zmm*) mutants show defects in homolog engagement that result in increased meiotic DSBs (Thacker *et al*, 2014).

We observe a parabolic relationship between crossover frequency and SNP density. Initially, increasing SNP density associates positively with crossovers, until a threshold is crossed and the relationship becomes negative. This is consistent with observations that very high local SNP density associates with crossover suppression at the fine scale in Arabidopsis, for instance when mapping crossovers within single crossover hotspots (Choi *et al*, 2016; Serra *et al*, 2018b). This parabolic relationship is likewise consistent with larger structural variation suppressing crossover formation in *Arabidopsis* (Rowan *et al*, 2019). Although *A. thaliana* is predominantly self-fertilizing, this species evolved from an outcrossing ancestor ~ 0.8–1.2 million years ago (Bomblies *et al*, 2010; The 1001 Genomes Consortium *et al*, 2016; Fulgione & Hancock, 2018). As self-fertilization causes an increase in homozygosity, the positive associations between sequence diversity and crossover frequency identified here may represent a means to bias recombination towards variable regions, in order to maximize the diversifying effects of meiosis. However, the possible drive to promote recombination at mismatched strand invasion events likely represents a trade-off against a higher risk of non-allelic crossover, the balancing of which would be particularly important in more repetitive genomes. For example, HR was found to increase in a tomato line carrying a chromosome substitution from a wild relative, when mismatch repair was disrupted (Tam *et al*, 2011). However, in this respect we observed equal suppression of crossovers within inversions in both wild type and *msh2*. Hence, it is possible that the effect of MSH2 on crossovers depends on the chromosome region, the level and type of polymorphism, genetic background and the juxtaposition with surrounding regions. Accordingly, the interaction of MutS mismatch sensors with meiotic recombination may be fine-tuned between species, according to genome structure, levels of outcrossing and diversity.

Plant species with large genomes, for example wheat, barley, maize and tomato, show extensive regions of crossover suppression surrounding the centromeres and heavily biased recombination towards the sub-telomeres (Higgins *et al*, 2012; Choulet *et al*, 2014; Li *et al*, 2015; Demirci *et al*, 2017). Crossover suppression in the centromere proximal regions correlates with the presence of dense heterochromatic modifications, including DNA methylation and H3K9me2 (Higgins *et al*, 2012; Choulet *et al*, 2014; Li *et al*, 2015; Demirci *et al*, 2017). The Arabidopsis centromere proximal regions are also heterochromatic and show high levels of DNA methylation, H3K9me2 and nucleosome occupancy, coincident with suppression of meiotic DSBs and crossovers (Yelina *et al*, 2012, 2015; Choi *et al*, 2018; Underwood *et al*, 2018; Walker *et al*, 2018; Lambing *et al*, 2020b). However, in Arabidopsis the physical extent of heterochromatin relative to euchromatin is reduced, compared to plant species with large genomes. It is noteworthy that despite plant species showing extensive variation in physical genome size and heterochromatin content, they typically experience ~ 1–2 crossovers per chromosome per meiosis (Mercier *et al*, 2015). As a consequence, the Arabidopsis genome shows a relatively high crossover frequency (~ 4–5 cM/Mb), compared to plants with large genomes (e.g. wheat ~ 0.1–0.2 cM/Mb) (Choulet *et al*, 2014; Serra *et al*, 2018b). Therefore, differences in the relative amounts of euchromatin and heterochromatin between species contribute to varying crossover landscapes along the telomere-centromere axes. It is important to note that Arabidopsis telomeres are observed to cluster with the nucleolus in early prophase I, in a bouquet configuration (Armstrong *et al*, 2001), which may relate to increased crossovers observed in distal regions in this species. Interestingly, Arabidopsis male meiosis shows elevated crossovers specifically in the sub-telomeric regions (Drouaud *et al*, 2007), which is dependent on the Class I repair pathway (Fernandes *et al*, 2017). Hence, genome size, chromosome number and heterochromatin content, in addition to polymorphism density, are likely to have significant effects on the recombination landscape between species.

The effects of *msh2* on Arabidopsis meiotic crossovers contrast with those observed in budding yeast (Borts & Haber, 1987; Alani *et al*, 1994; Chambers *et al*, 1996; Hunter *et al*, 1996; Chen & Jinks-Robertson, 1999; Martini *et al*, 2011; preprint: Cooper *et al*, 2018). Specifically, we observe no significant change in crossover number in Arabidopsis between *msh2* and wild type. In contrast, *msh2* crossovers increase by 1.2- to 1.4-fold per meiosis in budding yeast (Martini *et al*, 2011; preprint: Cooper *et al*, 2018). Furthermore, crossovers remodel to less diverse regions in Arabidopsis *msh2*, whereas in budding yeast the opposite is true, with crossovers remodelling to more diverse regions in *msh2* (preprint: Cooper *et al*, 2018). We propose that the differences in *msh2* phenotypes between species reflect differences in genome architecture and regulation of meiotic recombination. We note that the phenotypes of orthologous mutations of key regulators of meiotic recombination also differ between Arabidopsis and budding yeast. For example, *recq4a recq4b* in Arabidopsis shows a 3.3-fold increase in Class II crossovers and high fertility (Séguéla-Arnaud *et al*, 2016; Serra *et al*, 2018b). In

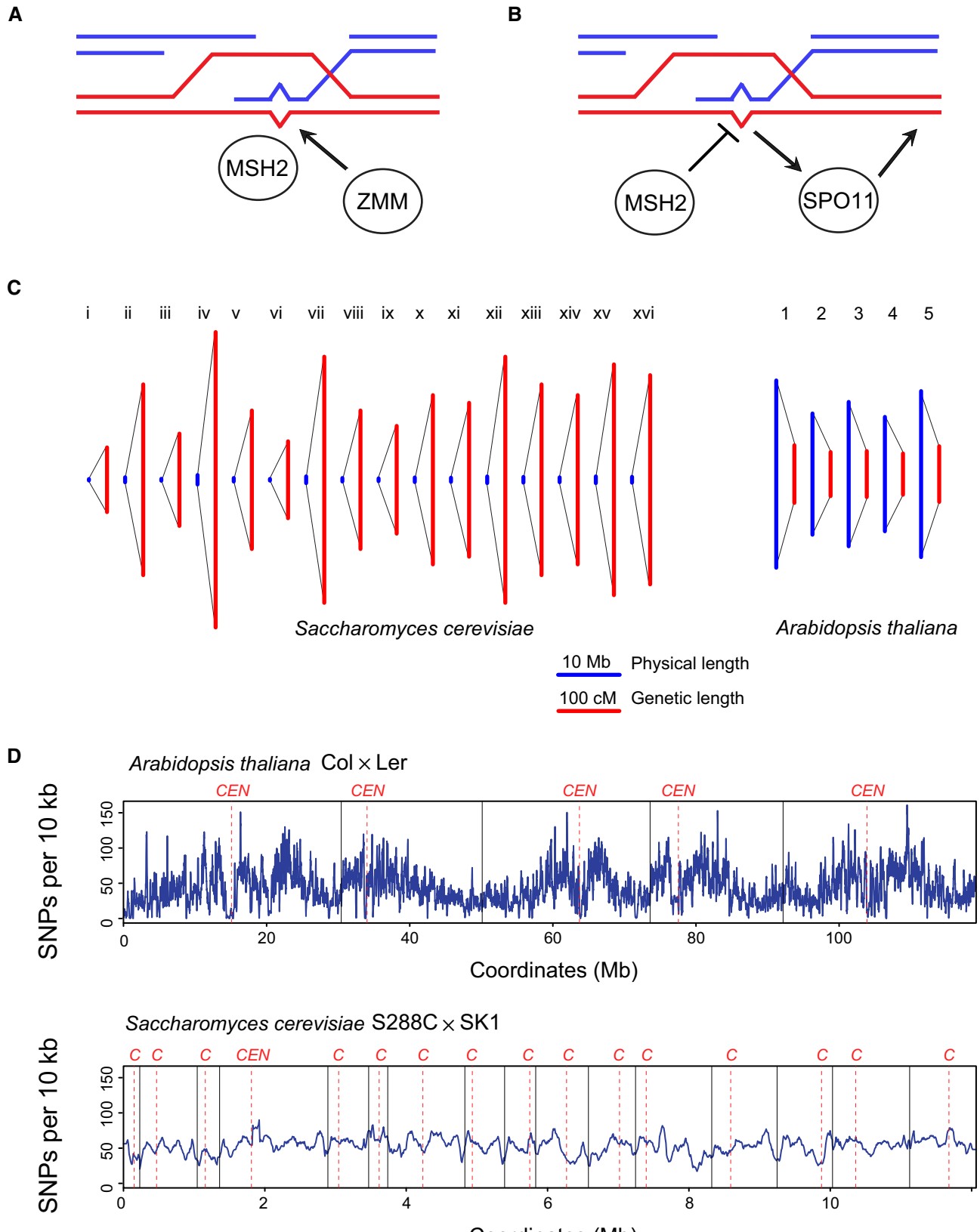

**Figure 8.**

**Figure 8. Models for regulation of meiotic recombination by MSH2 in *Arabidopsis* and budding yeast.**

A   Diagram representing interhomolog strand invasion between red and blue homologous chromosomes during meiotic prophase I. The resulting displacement loop contains a mismatch (bulge), which is recognized by a MSH2 MutS heterodimers. This causes recruitment or activation of the Class I (ZMM) repair pathway and increases the chance of crossover repair at, or in proximity to, the mismatched intermediate.

B   Diagram representing the same scenario as in A., but here MSH2 dependent recognition of the mismatch inhibits progression of recombination. This triggers feedback signalling that recruits SPO11-1 complexes to this region and increases meiotic DSB and crossover frequency.

C   Comparison of the budding yeast and Arabidopsis physical maps (blue) and genetic maps (red) (Mancera *et al*, 2008; Serra *et al*, 2018b), shown in megabases and centiMorgans, respectively.

D   Plots of SNP density per 10 kb along the chromosomes for Arabidopsis Col × Ler and budding yeast S288c × SK1 crosses (preprint: Cooper *et al*, 2018). Telomere positions are shown by vertical lines and centromeres by vertical dashed lines. Centromeres are also labelled above the plots in red, as "*CEN*" or "*C*".

contrast, budding yeast *sgs1* mutants accumulate aberrant joint molecules during meiosis and crossovers are either reduced or unchanged (Rockmill *et al*, 2003; Jessop *et al*, 2006; Oh *et al*, 2007; De Muyt *et al*, 2012; Zakharyevich *et al*, 2012).

Despite conservation of the Class I repair pathway between budding yeast and Arabidopsis, other aspects of genome organization are significantly different. For example, Arabidopsis has a larger and more repetitive genome (119.1 Mb over five chromosomes), compared with budding yeast (12.1 Mb over 16 chromosomes). Crossover frequency is far higher in budding yeast, with ~ 74 crossovers per meiosis in S288C/SK1 hybrids, compared to ~ 10 crossovers per meiosis for Arabidopsis Col/Ler hybrids (Fig 8C) (Wijnker *et al*, 2013; preprint: Cooper *et al*, 2018). However, both species are estimated to form a similar number of DSBs (~ 150–250) per meiosis (Buhler *et al*, 2007; Pan *et al*, 2011; Ferdous *et al*, 2012), which indicates that the anti-crossover pathways are likely more dominant in Arabidopsis. Additionally, Arabidopsis shows greater variation in polymorphism density along the chromosomes, compared with budding yeast (Fig 8D). This has the potential to cause differences in MSH2 recruitment between regions with varying levels of sequence polymorphism, which may cause feedback processes to emerge during prophase I.

Finally, we note that *msh2* meiotic recombination phenotypes differ between other species. For example, crossover frequency measured at varying scales in mice did not change in *msh2* or *pms2* (Qin *et al*, 2002; Kolas *et al*, 2005; Peterson *et al*, 2020). Mouse *msh2* mutants show heteroduplex retention in crossover products, but MSH2-dependent suppression of meiotic recombination was not observed (Peterson *et al*, 2020). Interestingly, this is in contrast to the anti-recombination role of MSH2 in mitotic cells observed during HR in both mouse and Arabidopsis (Elliott & Jasin, 2001; Emmanuel *et al*, 2006; Li *et al*, 2006). In *Caenorhabditis elegans*, mismatch repair was found to play roles in promoting heterologous meiotic recombination involving a large 8 Mb inversion (León-Ortiz *et al*, 2018). The *rtel1* mutant increases heterologous recombination within this inversion, which was suppressed by *msh2* (León-Ortiz *et al*, 2018), consistent with a pro-crossover role for MSH2 in this context. In *Schizosaccharomyces pombe*, *msh2* mutants show increased mitotic mutation rate, delayed meiotic progression, defective meiotic chromosome structure and a failure to undergo mating-type switching (Rudolph *et al*, 1999). Together, this shows that *msh2* meiotic recombination phenotypes are highly dependent on the species tested. Our data are consistent with MSH2 heterodimers acting as mismatch sensors that modulate meiotic recombination outcomes according to polymorphism density. However, we propose that the species, cell type, cell cycle stage and structure of

mismatched DNA may all influence the consequence of mismatch recognition by MSH2 heterodimers during HR.

# Materials and Methods

### Plant materials and growth conditions

The *msh2-1* T-DNA insertion line (SALK_002708) was obtained from the Nottingham *Arabidopsis* Stock Centre (NASC). Arabidopsis accessions Col, Ler, Bur, Ct, Ws and the CLC backgrounds were from our laboratory stocks. FTLs *I1b* (Francis *et al*, 2007), *5.10* (Wu *et al*, 2015) and *420* (Melamed-Bessudo *et al*, 2005) were kindly provided by Greg Copenhaver, Scott Poethig and Avraham Levy, respectively. The *HEI10-OE* line corresponds to transgenic line "C2", previously reported as "*HEI10*" (Ziolkowski *et al*, 2017; Serra *et al*, 2018b).

### Genotyping-by-sequencing library preparation

Genomic DNA was extracted using CTAB and used to prepare sequencing libraries, as described (Rowan *et al*, 2015). Briefly, 150 ng DNA per sample was digested with 0.3 units of dsDNA Shearase (Zymo Research) in a volume of 15 μl. The resulting DNA fragments were end-repaired with 3 units of T4 DNA polymerase (New England Biolabs), 10 units of T4 polynucleotide kinase (Thermo Fisher Scientific), 1.25 units of Klenow fragment (New England Biolabs) and 0.4 mM dNTPs, in a volume of 30 μl for 30 min at 20°C. DNA fragments were cleaned as described (Rowan *et al*, 2015), and the protocol was followed until the DNA fragment size selection step. To size-select DNA following Illumina barcoded adapter ligation, 30 μl of a mixture of eight concentrated DNA libraries were combined in a tube containing 48 μl of 1:1 AMPure XP magnetic SPRI beads:water (Beckman-Coulter). After 5 min incubation at room temperature, the samples were placed in a magnetic rack and allowed to clear before supernatant was transferred to a fresh tube and mixed with 0.12 volumes of undiluted SPRI beads. After 5 min at room temperature, the tubes were placed on a magnetic rack and allowed to clear. The supernatants were discarded, and the beads were washed twice with 80% ethanol. DNA was eluted in 20 μl of 10 mM Tris (pH 8.0). Twelve microliter of the eluate was used for PCR amplification in a reaction volume of 50 μl using KAPA HiFi Hot-Start ReadyMix PCR kit (Kapa Biosystems) and the reported DNA oligonucleotides (Rowan *et al*, 2015). Twelve cycles of PCR amplification were performed, and PCR products were then purified using SPRI beads and quantified using a

Bioanalyzer. The resulting libraries were subjected to paired-end 150 base pair sequencing on an Illumina NextSeq instrument, with 96 barcoded libraries sequenced per lane.

## Genotyping-by-sequencing bioinformatics analysis

After sequencing of each $F_2$ population, undemultiplexed data from 96 libraries were aligned to the TAIR10 genome assembly using Bowtie2. Single-nucleotide polymorphisms (SNPs) were called using SAMtools and BCFtools. SNPs were filtered to remove those with qualities < 100 and > 2.5 × mean coverage and additionally repeat masked. Data were then demultiplexed for each library and aligned to TAIR10 and analysed for genotypes at the previously identified SNPs. These data were then used to identity crossover sites using the TIGER pipeline (Rowan *et al*, 2015). For fine-scale analysis, crossovers with a resolution > 10 kb were filtered out. Following sequencing of the *msh2* libraries, we identified the presence of Col introgressions which had remained after backcrossing, including around the *msh2-1* T-DNA. These introgressions cause false crossover calls by TIGER so these were masked from analysis in *msh2* data and the corresponding wild type control. The masked regions for Col × Ler were Chr4: 0–170 kb and Chr4: 7.80–8.02 Mb. The masked regions for Col × CLC were Chr1: 3.46–4.65 Mb, Chr1: 11.30–11.33 Mb, Chr2: 4.47–5.39 Mb, Chr4: 16.12–18.88 Mb and Chr4: 3.13–4.80 Mb. The *msh2* sequencing data generated are available at ArrayExpress accession E-MTAB-8252.

To evaluate differences in crossover patterns between $F_2$ populations, crossovers were counted in 10 proportionally scaled windows ($10^{ths}$) between each telomere and centromere. For each population, windowed crossover frequencies were summed across all $F_2$ individuals and chromosome arms. For each $10^{th}$ of the combined chromosome arms, crossovers were modelled by Poisson regression with the log link function using the *glm* function in R, with population included as the predictor variable. Model goodness-of-fit was evaluated using chi-square tests based on the residual deviance and degrees of freedom ($P > 0.05$), by comparison of observed and model predicted means and standard errors, and by comparison of Bayesian information criterion values for Poisson and alternative regression models.

To analyse double crossovers (DCOs), we filtered for chromosomes showing parental-heterozygous-parental genotype transitions (e.g. Col-Het-Col or Ler-Het-Ler in Col × Ler $F_2$ individuals) (Drouaud *et al*, 2005; Lambing *et al*, 2020a) and recorded their distances. For each individual and chromosome, a matched set of randomly chosen points were generated and these distances recorded. For each population, 2,000 random data sets were generated. Permutation tests were then performed to assess the significance of differences between observed DCOs and random distances.

## Pollen-based FTL measurements of crossover frequency

Inflorescences were collected from plants hemizygous for FTL transgenes in a *cis* configuration ($RG/^{++}$) in 50 ml falcon tubes from mature plants. Pollen-sorting buffer (PSB; 10 mM $CaCl_2$, 1 mM KCl, 2 mM MES, 5% sucrose (w/v), 0.01% Triton X-100 (v/v), pH 6.5) was added, and the pollen extracted by vigorous shaking. The solution was filtered through a 40 μm cell strainer (Stemcell Technologies) into a fresh falcon tube and centrifuged at 450 *g* for 5 min at

4°C. The supernatant was gently discarded and the pellet washed with PSB (minus Triton X-100) and centrifuged at 450 *g* for 5 min at 4°C. The supernatant was discarded, and the pellet re-suspended in 600 μl PSB. Flow cytometry was performed on a BD Accuri C6 Flow Cytometer (BD Biosciences) or Guava easyCyte 8HT Cytometer (Millipore) equipped with a 488 nm laser and 530/30 and 570/20 nm band-pass filters. To select pollen by size, events were separated based on forward and side scatter. Hydrated pollen was gated to exclude dead or damaged material. Finally, events could be identified by emission signal in red (R3), yellow (R6), double-colour (R4) or non-colour (R5) categories. Crossover frequency (cM) was calculated as 100 × (R6/(R6 + R4)) (Yelina *et al*, 2013).

## Seed-based FTL measurements of crossover frequency

Seeds were collected from plants hemizygous for FTL transgenes in a *cis* configuration ($RG/^{++}$) and cleaned to remove debris. Seed images were captured with a Leica M165 FC dissecting epifluorescence microscope (Leica Microsystems), using bright field, UX + mCherry and UV + GFP filters. Images were processed through a CellProfiler pipeline, which identifies seed boundaries and assigns each object a fluorescent intensity value (Ziolkowski *et al*, 2015). Thresholds between fluorescent and non-fluorescent seed were set manually using fluorescence histograms. The number of seed in each class was used to calculate crossover frequency using the formula: cM = 100 × (1 − (1 − 2(R + G)/T)/2), where R is red only seeds, G is green only seeds and T is the total number of seeds.

## Alexander staining of pollen

Mature flowers were selected from inflorescences on the primary floral axis, and anthers were agitated in 20 μl of Alexander Stain Solution (0.01% malachite green, 10% ethanol, 0.05% acid fuchsin, 0.005% orange G, 4% glacial acetic acid, 25% glycerol) to release pollen grains. A cover slip was applied and sealed with rubber solution (Weldtite). Slides were incubated overnight at 37°C and screened for pollen viability using a standard bright field microscope.

## DAPI-stained meiotic chromosome spreads

Arabidopsis inflorescences were collected 6 weeks post-germination from the primary floral axis and fixed in 3:1 ethanol:acetic acid at 4°C. The fixative was replaced after 3 h and again after 12 h. Fixed inflorescences were dissected in fresh fixative using forceps under a stereomicroscope (Leica). Buds of length 0.2–0.7 mm were selected, which correspond to floral stages 8–10. The fixative was removed and the buds washed three times in 1 ml of Citrate Buffer (44.5 mM citric acid, 55.5 mM sodium citrate) for 2 min. To digest the cell walls, buds were incubated with 3.3 mg/ml cellulase (Sigma) and 3.3 mg/ml pectolyase (Sigma) diluted in Citrate Buffer in a moist box for 1 h 30 min at 37°C. The enzyme solution was removed and 1 ml of Citrate Buffer added. Individual buds were transferred to a drop of water on a glass slide and gently disrupted with a brass rod. Following this, two 5 μl drops of 60% acetic acid were added and the resulting solution was mixed with a needle and the slides incubated on a heat block at 48°C for 1 min. One hundred fifty

microliter of ice-cold fixative was applied to the slide, and the slide rocked from side-to-side to spread the mixture, followed by inversion and drying. Fourteen microliter of DAPI Solution (10 µg/ml; Sigma) diluted in VECTASHIELD was applied, and an inverted DMI6000 B microscope (Leica) used for image capture.

### Immunostaining meiotic chromosome spreads for MSH2 and MSH4

The MSH2 MutS III domain was amplified using Q5 DNA polymerase (NEB) from wheat (Cadenza) spike cDNA using primers MSH2F1 (AGCATATGCGACTTGATTCTGCCG), incorporating an *Nde*I site and MSH2R2 (AACTCGAGTTGGCAATCACCAGCAC), incorporating a *Xho*I site. PCR products were ligated into pDrive (Qiagen) and sequenced. The MSH2 insert was digested with *Nde*I/*Xho*I and ligated in-frame into pET21b (Merck) and transformed into *E. coli* BL21 expression cells (NEB). A 48 kDa MSH2 fragment protein was expressed, purified using nickel resin and refolded and used as an antigen to raise a rabbit polyclonal serum (DC Biosciences). Immunocytology was performed as previously described (Higgins *et al*, 2004). The following antibodies were used: α-ASY1 (rat, 1/500 dilution) (Armstrong *et al*, 2002), α-MSH2 (rabbit, 1/200 dilution) and α-MSH4 (Higgins *et al*, 2004) (rat, 1/500 dilution). Microscopy was carried out using a Ni-E Fluorescence Microscope (Nikon). Image capture, analysis and processing were conducted using NIS-Elements software (Nikon). A manual alignment tool in NIS-Elements was utilized for aligning images and the count tool for quantifying overlapping foci.

### CRISPR-Cas9 mutagenesis of *MSH2*

To obtain *msh2* mutant lines in backgrounds with varying Col/Ct heterozygosity, a pair of gRNAs targeted within exon four of *MSH2* were designed. Agrobacterium transformation was performed using a vector containing the *MSH2* gRNA pair under the *U3* and *U6* promoters, and a *ICU2::Cas9* transgene. Transformants were genotyped by PCR amplification with primers flanking the *MSH2* gRNA target sites, and Sanger sequencing was performed to detect deletions. Mutants with heritable deletions causing a frame shift in *MSH2*, and not carrying the CRISPR-Cas9 construct, were identified for further experiments. Recombinant lines with the desired patterns of heterozygosity were obtained from a cross between Col, with or without *msh2*, or *HEI10-OE* line C2, and lines carrying defined patterns of Col or Ct polymorphism (Appendix Figs S11 and S12).

## Data availability

The fastq DNA sequencing data from this publication have been deposited to the ArrayExpress database (www.ebi.ac.uk/arrayexpress) and assigned the accession identifiers E-MTAB-8252 (http://www.ebi.ac.uk/arrayexpress/experiments/E-MTAB-8252/), E-MTAB-9369 (http://www.ebi.ac.uk/arrayexpress/experiments/E-MTAB-9369/) and E-MTAB-9370 (http://www.ebi.ac.uk/arrayexpress/experiments/E-MTAB-9370).

**Expanded View** for this article is available online.

## Acknowledgements

We acknowledge support from a Sainsbury Ph.D studentship (ARB), BBSRC sLOLA BB/N002628/1 (SD, JD, IRH), ERC CoG grant SynthHotSpot (CAL, AJT, IRH), a BBSRC DTP studentship (EJL), Polish National Science Centre grants 2016/21/B/NZ2/01757 (NK, PAZ), 2016/22/E/NZ2/00455 (JD, TB, PAZ) and a Foundation for Polish Science grant POIR.04.04.00-00-5C0F/17-00 (MSL, PAZ). BR was supported by the Max Planck Society through funds to Detlef Weigel. We kindly thank Avraham Levy, Greg Copenhaver and Scott Poethig for providing FTLs. Monika Gazecka is acknowledged for technical assistance. We thank Timothy Cooper and Matt Neale for sharing a budding yeast S288c × SK1 polymorphism data set and for useful discussions.

## Author contributions

All authors designed research and experiments. ARB, JD, MS-L, SD, NK, CL, EJL, TB, BR and JDH performed experiments. ARB, JD, MS-L, SD, AJT, NK, CL, EJL, TB, BR, JDH, PAZ and IRH analysed data. ARB, PAZ and IRH wrote the paper with comments from all authors.

## Conflict of interest

The authors declare that they have no conflict of interest.

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
