## [Review Process File · The EMBO Journal]

MSH2 shapes the meiotic crossover landscape in relation to interhomolog polymorphism in *Arabidopsis*

Alexander Blackwell, Julia Dluzewska, Maja Szymanska-Lejman, Stuart Desjardins, Andrew Tock, Nadia Kbiri, Christophe Lambing, Emma Lawrence, Bieluszewski Tomasz, Beth Rowan, James Higgins, Piotr Ziółkowski, and Ian Henderson

DOI: [10.15252/embj.2020104858](https://doi.org/10.15252/embj.2020104858)

Corresponding author(s): Ian Henderson (irh25@cam.ac.uk) , Piotr Ziółkowski (pzio@amu.edu.pl)

Review Timeline:

Submission Date:	9th Mar 20
Editorial Decision:	15th Apr 20
Revision Received:	3rd Jul 20
Editorial Decision:	17th Jul 20
Revision Received:	13th Aug 20
Accepted:	19th Aug 20

Editor: Hartmut Vodermaier

Transaction Report:

Thank you for submitting your manuscript on MSH2 and meiotic crossover in Arabidopsis for our consideration. I have now received the comments from three reviewers, as well as some informal further input from an expert editorial advisor. In light of the overall positive feedback from these experts, we would be interested in pursuing this study further for EMBO Journal publication, pending satisfactory revision of several important interpretational and presentational issues.

As you will see from the reports copied below, the referees by and large appreciate the presented data and their quality, but disagree with many of the conclusions derived from them. Particular issues are the comparisons of local vs. global effects, the possibility that polymorphism density and crossovers may be less directly related to each other than to unidentified additional factors, the (general) involvement of Class 1 crossovers, and ambiguities related to male vs. female germline. All referees also note that some of the described effects are rather subtle, and call not just for altered presentation but also (computational) re-analyses of some of the data, with the notion that this may reveal more concrete insights into certain aspects of the studied process. Related to the underlying mechanisms, our editorial advisor also noted some previously reported observations on MSH2 that might be considered/integrated when discussing possible explanatory models. Finally, all three referees agree that parts of the statements in the current title and abstract are not warranted and would need to be revised.

Given that improving these issues should not necessarily require additional experimental work, we would be open to considering this work further in case you should be prepared to comprehensively address the key concerns through further data analyses as well as textual and presentational adaptations during a major revision of this study. Please let me know if you should have any questions/comments in this regard, which I would be happy to discuss with you at any time during such a revision. Since we realize the current difficulties with research work in the present COVID-19 pandemic situation, we would of course also be open to extending the duration of revisions, during which competing work appearing elsewhere would not affect our final assessment of your study.

Further information on preparing and uploading a revised manuscript can be found below and in our Guide to Authors. Thank you again for the opportunity to consider this work for The EMBO Journal, and I look forward to hearing from you.

Referee #1:

Blackwell et al. examine the relationship between polymorphism density and meiotic crossover distributions in several hybrid *Arabidopsis* strains. The basic finding is that, on a population basis, meiotic crossovers show a modest preference for regions that have greater sequence polymorphism (which are also centromere-proximal), and mutation of the MSH2 gene, required for mismatch recognition, flattens this distribution to greater or lesser extents, depending upon the hybrid, again arguing for an contribution of mismatch recognition to crossover location. Increasing Class 1 crossovers about 2.5-fold by overproducing HEI10 also flattens the crossover distribution, and increasing Class 2 crossovers (by *recq4a recq4b* mutation) to at least 2/3 of total crossovers results in a shift of crossovers to centromere-distal regions, leading authors to suggest that the mismatch-enriched crossovers are Class 1. Additional data document Msh2 recruitment to meiotic chromosomes, and attempt to recapitulate the effect using pairwise combinations of spore color markers. Overall, these data stand in contrast to what has been seen in budding yeast, where a roughly 2-fold greater polymorphism density causes a MSH2-dependent reduction in crossovers, and where the opposite shift upon *msh2* mutation is seen.

These intriguing findings are extensively documented and will serve as a resource for future experimentation, although overall the effects are modest and insight into why this phenomenon occurs is quite limited. Authors propose two models: one in which MSH2 complexes, enriched in regions of elevated heterozygosity, might recruit the MutLgamma complex that resolves intermediates as crossovers; the other, in which MSH2-mediated heteroduplex rejection delays homolog synapsis, leading to continued double-strand break formation and thus greater recombination event frequencies. Neither model is tested and others, of course, are possible.

Most of my concerns with the manuscript involve the presentation of the data and can be easily addressed by revising figures. However, there are two concerns regarding interpretation of the data which I would like to see addressed by substantial text revision.

1. The first involves the relationship between crossover and polymorphism density, which is weakly correlated, raising the question of if this correlation reflects a direct or indirect relationship (i.e. both correlated with a third factor). This concern is amplified by the observation that, on the "average arm" map (Figures 1c, 3c), polymorphism density seem to decrease almost linearly from centromere to telomere, whereas crossover density peaks near the centromere and then declines about 2-fold more rapidly to a plateau, so that in at least some hybrids there are chromosome segments with substantial change in polymorphism density that show no change in crossovers. It might be worth performing a multiple regression or principal component analysis that also considers position relative to the centromere, to determine if location or polymorphism density makes a greater contribution to crossover density.
2. The second involves the identification of the affected crossovers as Class 1. This is done by examining HEI10 overproducing (increases Class 1 crossovers) and RECQ4-deficient strains (increases Class 2 crossovers). While authors use a population average measure to SNP density around crossover sites to argue that mismatch-enriched crossovers are Class 1, the fact is that other data (Figure 3c, Figure S8) indicate that the correlation has been completely lost, and that the additional crossovers are, if anything, anticorrelated with SNP density. Basically, if the extra crossovers followed the same rules as those in wild type, one would expect that the pattern of crossovers across the chromosome would be preserved, as would be the correlation between SNP and crossover density. To the contrary, the distribution of crossovers along chromosome arms is

essentially flattened (excluding peri-centromere and peri-telomere effects; see Figure 3c), and the correlation coefficient drops from 0.37 to -0.38 (Figure S8). Basically, the data do not support the authors' conclusion, and it should be dropped from the title and from the rest of the paper.

3. The other concern I have is with regard to data presentation. In many of the figures, Y axes with non-zero roots are used to visually increase the apparent magnitude of what really are very small differences. These include: Figure 1c; Figure 2a, c; Figure 3c; Figure 4b; Figure 6c, d; crossover plots in Figure S4; Figure S5b, c; and crossover plots in Figure S11. In addition, the deviation from poisson distribution plots in Figure S1 and Figure S10 are presented in an unwieldy form, with columns rooted on the fitted curve rather than on the X axis, and including plotting the square root of frequency (rather than just frequency) on the Y axis. This has the effect of giving the tails of the distribution more visual impact and decreasing the visual contribution of the majority of events. The net effect of all of this is to visually increase the apparent magnitude of what are really quite modest differences and effects. If authors wish to retain this kind of presentation, they ought to at least present in supplementary material plots where the X axis intercepts the Y axis at 0, and where Y axes represent linear values.

Minor comments:

4. The analysis of motif association reveals only very weak correlations and adds nothing to the conclusions. Authors should consider removing it.

5. The FTL analysis does not appear to support a relationship between polymorphism density and crossovers or an effect of msh2 mutation, but it's hard to tell what would be expected on the basis of the crossover density maps in Figure S9a and b. Perhaps the addition to the plots in panels C and D of the area under the red and blue curves, transformed into centimorgans, would help in interpretation? Or would it reveal serious discrepancies in interval map distances measure with these two methods.

6. While unlikely, it is formally possible that DSB distributions are affected by MSH2 genotype. This should be discussed.

7. In many of the plots, X and Y axis values are in a font that is so small as to be unreadable.

Referee #2:

This is a very interesting study, of high quality and very exhaustive, on the effect of sequence polymorphism on homologous recombination in plants and on the role of MSH2 in determining the distribution of crossovers as a function of sequence divergence at the local level or at the whole chromosome level. The data presented here are novel and this topic is of general interest for geneticists working on sexually reproducing organisms.

This work comes in the background of apparently conflicting evidence in the field on this topic. On the one hand, evidence in the literature of a suppressor effect of sequence divergence on crossovers that is mediated by MSH2 and in this work, evidence for a pro-crossover effect of MSH2. While the work presented here is quite impressive, I feel that the authors should have made a greater effort to discuss if these two clashing views can be reconciled or if one of them is plain

wrong or maybe both are wrong - I elaborate on this below (point 5). I also raise a number of points that the authors should consider.

1- I am wondering if the TIGER pipeline could underestimate the number of crossovers in less divergent regions? Usually Markov-based algorithms are more effective at determining the transition from one parental type to the other when there is more divergence in sequence. The borders of the CO in such cases might be very large and might have been omitted from some of the analyses.

2- In larger genomes than Arabidopsis there is a very large portion of the chromosome around the centromere with almost no recombination. I am curious if the authors have any thoughts as to why pericentric region close to the centromere have the highest degree of CO, and why we see this only in Arabidopsis. Could this reflect pairing patterns starting at centromeres? That these regions are more polymorphic is the explanation of the authors, but then why this is not the case in other species. Please discuss that.

3- The authors extend previous works from their and other labs by showing that in all crosses COs are more frequent in promoters and terminators which are also more polymorphic than exons. Could it be the explanation for this local effect of recombination in the diverse region (which would have little to do with polymorphism itself but be an indirect effect of chromatin accessibility). What is shown locally at the bp level might be quite different from the phenomenon seen at the whole chromosome level.

4- Any insight on the effect of msh2 on the reduction of CO in pericentric regions (Fig 3)?

5- Figure 6 is an analysis of the effect of sequence divergence and MSH2 at the chromosomal scale. Both the effect of divergence in one chromosomal segment on CO and the effect of MSH2 are dependent on the rest of the chromosome. Remarkably the effect of msh2 on CO in the subtelomeric region of chromosome 3 is inverse depending on divergence in the rest of the genome (e.g. Fig 6C HET-HOM versus HOM-HET). This is a very interesting finding but here I am really wondering if one can think of a mechanism that would explain both the local effect and the chromosomal effect-seems like two different phenomena? Please elaborate on that. It seems to me that the authors have not provided any satisfactory mechanistic model that would explain both phenomena.

I also think that the title is not reflecting the authors results. I would say that "the effect of MSH2 on CO in one given region is dependent on the sequence divergence surrounding this region" is a more appropriate description of the authors findings

Presenting things that way could explain discrepancies in conclusions with other works on the same subject: I note that the results with the 420 FTL are similar to what was reported by Emmanuel et al. 2006 who showed an increase in meiotic CO in msh2 in the HET-HET situation compared to HOM-HOM and reached the conclusion that MSH2 has an anti-recombination effect when looking at a specific interval in a specific genetic background configuration.

I think this is the main story of this work, that the MSH2 effect can be either pro or anti recombination depending on the flanking region heterozygosity. This is what the authors show us and this might settle a long-held controversy. For some reason they decided to have a simple bottom line that ignores their own results.

At the gene-scale I have the feeling that what we see is simply due to the strong pro-recombination effect of promoters which are nucleosome depleted, AT rich etc.. and happen to be more polymorphic than exons. At the chromosomal scale we have a much more puzzling story. I am

not sure if the authors have enough data but if they could find gene-free regions that undergo CO, they could use it as a control to show that this is polymorphism per se and not gene features that determine CO position. They could be in for some surprises and find that polymorphism has an anti-recombination effect when in non-gene regions.

To summarize, I find this work to be very interesting and I laud the authors for tackling a very difficult topic. I do not ask for new data, but for a reexamination of their results and of their interpretation as described above.

Referee #3:

In this paper Blackwell and co-workers investigate the relationship between sequence polymorphism and meiotic crossover frequency in *Arabidopsis thaliana* and they investigate the role of the mismatch repair protein MSH2 in meiotic recombination.

The authors first studied the correlation between CO frequency and SNP density in a number of genetic contexts (four different accessions and several mutant backgrounds) by mapping COs after DNA sequencing of F2 populations. Then they studied the effect of the MSH2 protein (a demonstrated component of the mismatch repair machinery) on meiotic recombination patterns and provide information on the dynamics of this protein during meiosis, based on immunofluorescence studies.

This paper represents a very nice piece of work, with lot of interesting and globally robust data; data are clearly presented and the paper is extremely well written. While I agree with many of the provided conclusions, I question the ones that concern the link between sequence polymorphisms and CO levels. These are too often overstated, starting with the title of the paper itself. My second concern, is that most of the data analyzed here have been generated based on F2 population sequencing. In consequence, recombination rates and profiles analysed correspond to an addition of male and female recombination. This is a problem for two things. First, it is well known that male and female meiosis have very contrasted patterns of recombination both in terms of rates and patterns (Giraut et al 2011). Second, it is also known that this sexual dimorphism is abolished in mutants such as *recq4ab* (Fernandes et al 2018), a genetic context used here to demonstrate that Class II COs are polymorphism insensitive. Therefore, I am afraid that many of the correlation data could be considerably biased in a sense that is difficult to estimate. In consequence I suggest that the authors keep the extrapolation of their results to establish a link between CO rates and SNP density to the discussion only.

More detailed comments

- Studies at global (200kb) scale: The authors do demonstrate that there is a correlation between the degree of polymorphisms and the rate of COs but they shouldn't hide/minimize the fact that the correlation is extremely weak. Besides, even if CO frequency is the highest in pericentromeric regions where SNP density is the strongest, it is not true in subtelomeric regions (lowest SNP density for high CO density). Therefore, I think that the conclusion of this paragraph (1196-1197) is misleading and does not take all the data into account.

Even if I agree that the authors revealed a correlation between crossover rates and high SNP density, the correlation could be easily biased by the fact that the highest SNP density is found in

pericentromeric regions which are well known to be different from the other parts of the chromosomes for many other things than SNP densities (recombination dynamics, DNA and chromatin marks, repeated element composition etc...). If high CO rates in these regions is driven by any of these parameter (and not polymorphisms), even partially, it will introduce a bias toward the strongest polymorphic regions. This should be considered to analyse the results, to draw conclusions and in the discussion.

-Studies at local scale (a few kbs)

Here the authors analyzed SNP density in windows of increasing sizes surrounding the detected COs. They detected a very clear enrichment in SNPs around COs. However, I think it would be interesting to analyze separately COs that occur in pericentromeres from the other ones. If the enrichment in SNP is not true for COs outside the pericentromeric regions it means that the correlation observed between SNP density and CO levels is biased by the elevated rates of recombination in pericentromeres which happens to be SNP rich. If it is the case, the conclusion of the paragraph (l214-215) as well as the general conclusions of the manuscript should be modified (see also my remark above).

- Then the authors perform the same kind of analyses in genetic backgrounds showing an increase in either Class I (HEI10 overexpressor) or Class II (recq4ab) or both (HEI10 recq4ab) mutants. While high SNP density is associated with COs in wt and HEI10 background, it is not the case anymore neither for recq4ab nor for HEI10 recq4ab backgrounds. the authors conclude that these data are consistent with class I COs that associate with regions of strong polymorphisms but they do not explain why no difference is observed between the recq4ab and HEI10 recq4ab genotypes. In addition, how do they explain the shape of the curve (fig2C) in recq4ab (and HEI10 recq4ab) that deviates from the random situation for large window sizes? Once again it would be interesting to test separately COs in pericentromeric regions from the others to see if both show an enrichment in SNP density only when Class I COs predominate. And here too the question of ignoring the different recombination profiles of male and female meiosis is a big issue, notably considering that it has been shown that recQ4ab mutations impact differently male and female meiosis (Fernandes et al 2018).

- msh2 mutant analyses

The change in recombination profiles observed in the msh2 F2 populations is very convincing, with a tendency to relocate COs from pericentromeric to the arms. However, I didn't find the conclusions drawn out of the analyses at a local scale convincing. The authors observed a reduced SNP enrichment at CO sites in msh2 only for the Col/Ler population. For the two others, the decrease is very mild (Col/Ct and Col/CLC). therefore, the conclusion drawn l338 and 339 is very surprising: "This is consistent with msh2 crossovers forming in regions of elevated meiotic DSBs, but with a reduced association with higher SNP density relative to wild type." This does not fit with the data.

-The FTL data are a bit confusing which is a pity because they should be a way to analyze male recombination independently from female. Why did the authors choose the 510 interval that indeed encompass Chr5 centromere but also largely extends over chromosome arm? Here it is clear that a better description of the regions denominated as "pericentromeres" should be provided (and not only refer to Choi et al). As for the l1b interval, the huge increase observed in Col/CLC background is difficult to understand. The male /female interpretation is poorly convincing because it is expected to act on both wild type and msh2. Here too the conclusion of the paragraph is not correct and largely overstated (l308/309)

- The MSH2 immunofluorescence study is a bit superficial. From what I can see from figure 5, MSH2 is more detected as a thread-like signal than as foci. In consequence, it is not surprising that MSH4 foci mostly colocalize with the MSH2 signal. If authors are convinced that MSH2 form foci, then they should provide better pictures and quantification as well as proper colocalization studies (counting, comparison to random colocalisation etc...). Beside a correct staging of meiocytes for which MSH2 signal is detected should be provided (using a marker that allows unambiguous staging of the meiocytes such as ZYP1).

Minor comments

- Abstract and title should be revisited except if the authors provide additional analyses that show a clear causal relationship between sequence polymorphism and CO rates.
- I wouldn't use "HEI10" for the HEI10 overexpressor line (it can be confused with the HEI10 gene)
- Previous published data on *A. thaliana* MSH2 should be better acknowledged, in the introduction as in the discussion.
- Explain better the CLC background
- A better description of the CrispCas9 msh2 allele (msh2-3 in Ct) should be provided.
- A clear definition of the pericentromeres is given l186 and reference is made to Choi et al. But it would be important to give the detailed coordinates on the chromosomes of the regions called pericentromeres and show their position on all the figures.
- Figure S3 left part is not clear: what is the correlation looked at?
- Table S8: did the authors checked for correct segregations of their markers? Some crosses appear to show a very strong bias among reciprocal phenotypic classes
- The discussion should provide a deeper examination of several points 1) divergence between male and female recombination rates 2) high recomb rates in subtelomeric regions of the chromosomes 3) consider other possibilities than a causal effect of SNP polymorphisms on preferential CO location

Editorial Advisor's comments (excerpt):

... What could further be considered is that MSH2 in plants has already been characterised to form heterodimers (in vitro) with MSH3, MSH6 and MSH7 (a plant specific MSH6 prologue) (Culligan et al 2000). This is interesting, because the Alani lab showed in 2014 (Rogacheva et al.) that yeast MSH2 / MSH3 heterodimers can stimulate the nuclease activity of the ZMM protein complex MLH1/MLH3 in vitro and thereby act as a pro-CO factor in theory. All this is not discussed in the paper but would certainly substantiate the claims of the authors. A model would place MSH2 (plus a partner protein like MSH3) at mismatched bases in meiotic recombination intermediates recruiting the MLH1/3 heterodimer to promote CO formation. ...

EMBOJ-2020-104858 – Response to reviewers

Referee #1:

Blackwell et al. examine the relationship between polymorphism density and meiotic crossover distributions in several hybrid Arabidopsis strains. The basic finding is that, on a population basis, meiotic crossovers show a modest preference for regions that have greater sequence polymorphism (which are also centromere-proximal), and mutation of the MSH2 gene, required for mismatch recognition, flattens this distribution to greater or lesser extents, depending upon the hybrid, again arguing for a contribution of mismatch recognition to crossover location. Increasing Class 1 crossovers about 2.5-fold by overproducing HEI10 also flattens the crossover distribution, and increasing Class 2 crossovers (by *recq4a recq4b* mutation) to at least 2/3 of total crossovers results in a shift of crossovers to centromere-distal regions, leading authors to suggest that the mismatch-enriched crossovers are Class 1. Additional data document Msh2 recruitment to meiotic chromosomes, and attempt to recapitulate the effect using pairwise combinations of spore color markers. Overall, these data stand in contrast to what has been seen in budding yeast, where a roughly 2-fold greater polymorphism density causes a MSH2-dependent reduction in crossovers, and where the opposite shift upon *msh2* mutation is seen.

We thank the reviewer for their accurate assessment of our work and key findings.

We acknowledge that the *msh2* results we observe in Arabidopsis are in contrast to *msh2* meiotic recombination phenotypes reported in budding yeast (e.g. Cooper et al., 2018; Hunter et al., 1996; Borts and Haber, 1987; Martini et al., 2011; Chen and Jinks-Robertson, 1999; Alani et al., 1994; Chambers et al., 1996).

Specifically, we observe that,

- (i) In Arabidopsis no significant change in total crossover number occurs in *msh2*, compared to wild type. In contrast, crossover numbers significantly increase by 1.2-1.4-fold per meiosis in budding yeast *msh2* (Cooper et al., 2018; Martini et al., 2011).
- (ii) Crossovers remodel to less diverse regions in Arabidopsis *msh2*. In budding yeast the opposite is true, with meiotic crossovers remodelling to more diverse regions in *msh2* (Cooper et al., 2018).

We propose that differences in *msh2* phenotypes between species reflects varying genome architecture and regulation of meiotic recombination. We note that the phenotypes of orthologous mutations in other regulators of meiotic recombination also differ between Arabidopsis and budding yeast. For example, *recq4a recq4b* in Arabidopsis shows a 3.3-fold increase in Class II crossovers and high fertility (Séguéla-Arnaud et al., 2016; Serra et al., 2018b). Whereas in budding yeast *sgs1* mutants, crossovers show a 1.6-fold decrease (Rockmill et al., 2003; Jessop et al., 2006b), and aberrant joint molecules accumulate in late prophase I that reduce fertility (Zakharyevich et al., 2012; Oh et al., 2007; Rockmill et al., 2003; De Muyt et al., 2012).

To compare the Arabidopsis *msh2* recombination phenotype with those observed in other species, we have added further material and references to the Discussion. We have moved Figure S13 to main Figure 8 to illustrate these differences more prominently. In this figure we compare the Arabidopsis and budding genomes in terms of physical and genetic map length and SNP density. We draw attention to the fact that far higher cM/Mb values are observed on the smaller budding yeast chromosomes, despite DSB estimates per nucleus being similar (150-250) (Buhler et al., 2007; Ferdous et al., 2012; Pan et al., 2011). This means a higher proportion of DSBs are repaired via non-crossover or inter-sister pathways in Arabidopsis, compared to budding yeast. Differences in the balance of meiotic repair pathways may have significance for the manifestation of *msh2* phenotypes between these species.

Arabidopsis also shows greater variation in SNP frequency along the chromosome telomere-centromere axes than observed in budding yeast (Fig. 8D). Notably the Arabidopsis pericentromeric regions show pronounced elevation in SNP density compared to budding yeast. These gradients of SNP frequency have the potential to trigger feedback processes to a greater degree than in budding yeast. Furthermore, Arabidopsis centromeres are highly repetitive and densely modified with heterochromatic marks (e.g. DNA methylation and H3K9me2), which suppress meiotic recombination. In contrast, SNP density is more uniform across the budding yeast centromere regions, which consist of single CENP-A nucleosomes that are not flanked by significant heterochromatin. As chromatin strongly influences recombination, we believe these differences are likely to be significant when comparing Arabidopsis and budding yeast.

Finally, we note that *msh2* meiotic recombination phenotypes differ between other species. For example, crossover frequency measured at varying scales in mice does not change in *msh2* or *pms2* (Peterson et al., 2020; Qin et al., 2002). For example, a recent analysis of strand exchange during meiotic recombination utilized the mouse *msh2* mutant to facilitate the identification of heteroduplex DNA (Peterson et al., 2020). Although heteroduplexes were retained in crossover products, this study found no evidence for MSH2-dependent suppression of meiotic recombination between divergent sequences (Peterson et al., 2020). This is in contrast to the anti-recombination role of MSH2 in mitotic cells observed in both mouse and Arabidopsis (Emmanuel et al., 2006; Li et al., 2006; Elliott and Jasin, 2001). In *C. elegans*, MMR was found to play an unexpected role in mediating heterologous meiotic recombination involving a large 8 Mb inversion (Leon-Ortiz et al., 2018). The *rtel1* mutant increases heterologous recombination within this inversion, which was suppressed by *msh2* (Leon-Ortiz et al., 2018), consistent with a pro-crossover role for MSH2 in this context. In *S.pombe* the *msh2* mutant shows increased mitotic mutation rate, a delay in meiotic progression, defective meiotic chromosomes and a failure to undergo mating-type switching (Rudolph et al., 1999). Together these studies demonstrate that *msh2* meiotic recombination phenotypes are highly dependent on the species tested. Further work will be required to understand which differences in genomic architecture or meiotic recombination pathways contribute to varying *msh2* recombination outcomes in different species.

These intriguing findings are extensively documented and will serve as a resource for future experimentation, although overall the effects are modest and insight into why this phenomenon occurs is quite limited. Authors propose two models: one in which MSH2 complexes, enriched in regions of elevated heterozygosity, might recruit the MutLgamma complex that resolves intermediates as crossovers; the other, in which MSH2-mediated heteroduplex rejection delays homolog synapsis, leading to continued double-strand break formation and thus greater recombination event frequencies. Neither model is tested and others, of course, are possible.

In the Discussion and Figures 8A and 8B we present two models to explain our observations of meiotic recombination in *msh2*. In each model we propose the biochemical function of MSH2 complexes is the same. Specifically, MSH2 heterodimers bind and recognize base pair mismatches occurring within joint molecules that arise following interhomolog strand invasion. We note that these models are not mutually exclusive.

- (i) In Figure 8A we present a model where MSH2 heterodimers recruit, stabilize or promote Class I activity at mismatched interhomolog strand invasion sites, increasing the chance of crossover repair. This model is supported by our genetic analysis, which implicates Class I repair in promotion of crossovers in polymorphic regions. *In vitro* biochemical data supporting this model comes from budding yeast, where Msh2/Msh3 heterodimers can directly stimulate the endonuclease activity of Mlh1/Mlh3 heterodimers (Rogacheva et al., 2014), and Mlh1/Mlh3 act within the Class I pathway.
- (ii) In Figure 8B MSH2 heterodimers bind to mismatched joint molecules and inhibit or slow further steps in recombination and repair. This slowed

progression stimulates feedback signaling to the DSB machinery, which promotes SPO11-1 recruitment to the same region, leading to a higher chance of crossover formation. A plausible pathway for feedback to the DSB machinery involves the ATM/ATR kinases (Carballo et al., 2008; Zhang et al., 2011; Kurzbauer et al., 2012; Garcia et al., 2003; Lange et al., 2011). Evidence for DSB feedback pathways exists in budding yeast, where mutants in the ZMM pathway that slow homolog engagement and recombination cause increased meiotic DSBs (Thacker et al., 2014).

Most of my concerns with the manuscript involve the presentation of the data and can be easily addressed by revising figures.

We have revised the manuscript and figures as suggested.

However, there are two concerns regarding interpretation of the data which I would like to see addressed by substantial text revision.

- 1. The first involves the relationship between crossover and polymorphism density, which is weakly correlated, raising the question of if this correlation reflects a direct or indirect relationship (i.e. both correlated with a third factor). This concern is amplified by the observation that, on the "average arm" map (Figures 1c, 3c), polymorphism density seem to decrease almost linearly from centromere to telomere, whereas crossover density peaks near the centromere and then declines about 2-fold more rapidly to a plateau, so that in at least some hybrids there are chromosome segments with substantial change in polymorphism density that show no change in crossovers. It might be worth performing a multiple regression or principal component analysis that also considers position relative to the centromere, to determine if location or polymorphism density makes a greater contribution to crossover density.**

We agree with the reviewer that multiple factors show transitions along the centromere-telomere axes of the chromosomes that have significance for the crossover landscape. For example, we have previously investigated the role of chromatin in shaping recombination frequency (Choi et al., 2018; Underwood et al., 2018; Lambing et al., 2020; Yelina et al., 2015). The centromeric and pericentromeric regions in Arabidopsis are heterochromatic and show high levels of DNA methylation, H3K9me2 and nucleosome occupancy, as well as suppressed DSBs and crossovers. Hence, despite these regions containing substantial polymorphism, the effect of chromatin dominantly suppresses recombination. To illustrate these points we have added a new Figure S2 that compares Col/Ler crossover and SNP frequency along the genome with published chromatin and recombination maps, including SPO11-1-oligos, nucleosomes (MNase-seq), H3K9me2, DNA methylation, H3K4me3 and gene density (Choi et al., 2018, 2016; Lambing et al., 2020; Yelina et al., 2015). We have discussed these patterns in the main text and discussion.

To quantitatively model the effects of multiple parameters on crossover occurrence, including SNP density and chromatin, we used a generalized linear model (GLM), also known as a multivariate regression model. These results are presented in the new Figure 2B and Table S5. This is an extension of a previous model where we examined the relationship between crossovers, meiotic DSBs (SPO11-1-oligos) and chromatin (Choi et al., 2018 *Genome Res*). In these models we consider all SNP intervals where it is possible to detect a crossover, corresponding to 534,780 intervals with a mean width of 224 bp. The binary response variable in the model is whether at least one crossover was observed in a given SNP interval, using a set of 3,320 crossovers mapped in Col/Ler F₂ plants (Serra et al., 2018). We then calculated explanatory variables for the same intervals, including SPO11-1-oligos, nucleosomes (MNase-seq), H3K4me3, DNA methylation and SNP density. For SNP density we calculated a rolling average of SNPs per kb using a 1 bp step and used these values to calculate mean SNPs/kb per interval. Data were then modeled with the glm2 function in R, using the binomial family with the

logit link function. The initial formula used was:

$$\text{Crossovers} \sim (\text{SPO11-1} + \text{nucleosomes} + \text{H3K4me3} + \text{DNA methylation} + \text{SNPs/kb} + \text{width})^2$$

For model selection we used the `stepAIC` function from the MASS package in R, using both forward and backward directions in order to minimize the Akaike Information Criterion (AIC). The formula for the final model was selected based on lowest AIC and includes significant interactions between parameters. Our model shows negative effects for nucleosomes and DNA methylation on crossovers, and positive effects for SPO11-1-oligos. Interestingly, we observe that SNPs/kb shows a parabolic relationship with crossovers (Fig. 2B). Initially, there is a positive relationship with higher SNPs/kb increasing the chance of observing a crossover. However, beyond a threshold, the relationship becomes negative. This is also reflected in correlations crossover frequency and polymorphism density (Fig. 2A). Together we interpret this as reflecting that SNP density has a positive effect on crossover formation, until high polymorphism density is reached, at which point it becomes inhibitory. We note that this is consistent with the suppressive effect of structural polymorphism on crossovers genome-wide (Rowan et al., 2019), and that within the *RAC1* resistance gene higher SNP density locally associates with suppressed crossover formation at the fine scale (Serra et al., 2018a). We have added discussion of the model to the paper and included the model outputs in main Figure 2 and Table S5.

- 2. The second involves the identification of the affected crossovers as Class 1. This is done by examining HEI10 overproducing (increases Class 1 crossovers) and RECQ4-deficient strains (increases Class 2 crossovers). While authors use a population average measure to SNP density around crossover sites to argue that mismatch-enriched crossovers are Class 1, the fact is that other data (Figure 3c, Figure S8) indicate that the correlation has been completely lost, and that the additional crossovers are, if anything, anticorrelated with SNP density. Basically, if the extra crossovers followed the same rules as those in wild type, one would expect that the pattern of crossovers across the chromosome would be preserved, as would be the correlation between SNP and crossover density. To the contrary, the distribution of crossovers along chromosome arms is essentially flattened (excluding peri-centromere and peri-telomere effects; see Figure 3c), and the correlation coefficient drops from 0.37 to -0.38 (Figure S8). Basically, the data do not support the authors' conclusion, and it should be dropped from the title and from the rest of the paper.**

The reviewer questions why the correlation between crossovers and SNPs decreases in *HEI10-OE*, despite our claim the Class I crossovers are associated with regions of higher SNP density. Our model proposes that Class I crossovers are promoted within regions of higher SNP density in wild type. Therefore, one prediction is that the additional *HEI10-OE* Class I crossovers would cause the correlation between crossovers and SNPs to increase relative to wild type, or stay the same.

In the process of revising the paper we have further refined our analysis of SNP and crossover correlations. We calculate crossovers and SNP density in 100 kilobase (kb) adjacent windows, with crossovers normalised by the number of F₂ individuals analysed. Centromeric regions were excluded from analysis, as defined in Underwood et al. 2018, due to the dominant effect of heterochromatin on recombination in these regions. Windows were grouped into percentiles ranked by SNP density, and SNP and crossover density were then calculated. Spearman's rank correlation coefficient was calculated between these values and plotted. Trend lines were fitted in ggplot using a generalized additive model (GAM) with formula = $y \sim \text{poly}(x, 2)$ (Fig. 2 and S7). In this analysis we observe a significant positive correlation between crossovers and SNPs in wild type Col/Ler ($r = 0.545$). In contrast, a strong significant negative correlation is observed in *recq4a recq4b* ($r = -0.772$) (Fig. S7). In *HEI10-OE* we observed that the correlation between SNPs and crossovers is not significant (Fig. S7).

To explain the decrease in the strength of correlation between SNP density and crossover frequency in *HEI10-OE*, we refer to recent work investigating Class I and Class II repair and sex differences in meiosis. In *Arabidopsis*, the sub-telomeric regions increase crossover frequency during male meiosis compared to female, which is dependent on the Class I pathway (Fernandes et al., 2017). As *HEI10* functions in the Class I pathway (Chelysheva et al., 2012; Ziolkowski et al., 2017; Serra et al., 2018b), we believe that *HEI10-OE* drives sub-telomeric recombination, similar to male meiosis. For example, this effect is evident in Figure 4C. As the sub-telomeric regions have lower SNP density, this is reflected in the weakened correlation between crossover and diversity in *HEI10-OE*. We also note that our fine-scale analysis of SNP density around crossover sites in *HEI10-OE* reveals similar enrichment of SNPs/kb relative to a random control, as observed in wild type (Fig. 3C). In contrast, in *recq4a recq4b* where the majority of crossovers occur via the Class II pathway, a significant reduction in SNPs/kb enrichment around crossovers is observed (Fig. 3C), in addition to a strong negative correlation between SNP density and crossover frequency at the genome scale (Fig. S7B).

We note that the juxtaposition effect is preserved in the *HEI10-OE* background (Fig. 7D). This is a further context where polymorphism promotes crossover in *HEI10-OE*. We refer to previous work where we tested the juxtaposition effect in *zip4* mutants and showed that it is absent, and is therefore Class I dependent (Ziolkowski et al., 2015). In this study we provide further evidence that Class I crossovers are positively associated with SNPs genome-wide, and that *MSH2* contributes to this relationship.

As suggested by the reviewer we have modified the title of the paper to remove mention of Class I. The revised title is 'MSH2 promotes meiotic crossover formation in regions of high sequence diversity in *Arabidopsis*'.

We note that in response to reviewer 3, we have renamed the *HEI10* genotype as *HEI10-overexpressor (HEI10-OE)*.

3. The other concern I have is with regard to data presentation. In many of the figures, Y axes with non-zero roots are used to visually increase the apparent magnitude of what really are very small differences. These include: Figure 1c; Figure 2a, c; Figure 3c; Figure 4b; Figure 6c, d; crossover plots in Figure S4; Figure S5b, c; and crossover plots in Figure S11.

We discuss each of these figures in turn.

Figure 1C shows crossover and SNP frequency along the chromosome telomere-centromere axes. We have regenerated these plots starting the y axis at zero.

Figure 2A and 2C (now Figure 3A and 3C) show SNPs/kb values around crossovers (red) versus random (blue) coordinates using increasing window sizes. We have regenerated the plots in A and C to start at zero on the y axis.

Figure 3C (now Figure 4C) shows plots of crossovers and SNPs along the telomere-centromere axes of the chromosome arms. We have regenerated these plots with the y axis starting at zero as requested.

Figure 4B (now Figure 5B) shows SNPs/kb in windows of the indicated physical sizes around crossovers versus random positions, in wild type compared with *msh2*. We have regenerated these plots starting the y axis at zero.

Figure 6C (now Figure 7C) shows 420 crossover frequency data in wild type *msh2* and *HEI10-OE* genotypes. We have regenerated these plots starting the y axis at zero.

Figures S4 and S11 (now Figure S10) analyse two types of data in windows around crossover

midpoints, compared to the same number of random positions. The first dataset is normalized enrichment of SPO11-1-oligos or MNase-seq data, calculated as $\log_2(\text{data}/\text{input genomic DNA})$. The second dataset examines DNA base frequency (AT versus GC) across the same windows. For these analyses the comparison between crossovers and random is the most relevant, rather than the position of the plots relative to zero. The random analysis provides an estimate of the genome-wide average for each dataset (which are non-zero), with which departures in the crossover plots can be compared.

Figure S5B - this figure presented analysis of DNA motifs associated with crossovers. As the reviewer's request (minor point 4) we have now removed this analysis and previous Fig. S5 from the manuscript.

In addition, the deviation from Poisson distribution plots in Figure S1 and Figure S10 are presented in an unwieldy form, with columns rooted on the fitted curve rather than on the X axis, and including plotting the square root of frequency (rather than just frequency) on the Y axis. This has the effect of giving the tails of the distribution more visual impact and decreasing the visual contribution of the majority of events. The net effect of all of this is to visually increase the apparent magnitude of what are really quite modest differences and effects. If authors wish to retain this kind of presentation, they ought to at least present in supplementary material plots where the X axis intercepts the Y axis at 0, and where Y axes represent linear values.

We thank the reviewer for this suggestion. We have regenerated these plots as requested by the reviewer, and incorporated them into Fig. 1A and Fig. 4A. The plots show histograms of observed (blue) crossovers per F_2 individual compared to the Poisson expectation (red). The observed crossovers in each of these populations are significantly different from the Poisson expectation, as assessed using goodness-of-fit tests. The previous plots in Fig. S1 and S10 have now been removed.

Minor comments:

4. The analysis of motif association reveals only very weak correlations and adds nothing to the conclusions. Authors should consider removing it.

We have followed the reviewer's suggestion and removed this analysis from the paper, including the associated supplemental table, figure and discussion in the main text.

5. The FTL analysis does not appear to support a relationship between polymorphism density and crossovers or an effect of *msh2* mutation, but it's hard to tell what would be expected on the basis of the crossover density maps in Figure S9a and b. Perhaps the addition to the plots in panels C and D of the area under the red and blue curves, transformed into centimorgans, would help in interpretation? Or would it reveal serious discrepancies in interval map distances measure with these two methods.

In our manuscript we presented data using two FTL crossover reporter intervals: (i) the distal sub-telomeric *11b* interval on chromosome 1, and (ii) the *5.10* interval that spans the centromere of chromosome 5, in addition to including adjacent chromosome arm sequence. We note that our *11b* data reports crossover frequency only in male meiosis, whilst *5.10* reports an average of crossover frequency in male and female meiosis. Within both intervals, we observe no significant difference in crossover frequency between wild type and *msh2* in inbred Col/Col backgrounds. This observation indicates that the effect of *msh2* on meiotic recombination in Arabidopsis requires the presence of interhomolog polymorphism. For example, crossover frequency increases within *11b* in *msh2* in both Col/Ler and Col/CLC hybrids. Conversely, crossover frequency significantly decreases in *msh2* within the centromere-spanning *5.10* interval in Col/Ler and Col/CLC hybrids. We also note that crossovers increased in the *msh2* mutant compared to wild type within the distal 420 FTL interval in a Col/Ct hybrid background (Fig. 7C). Together

these measurements are consistent with the crossover redistribution phenotype observed in our GBS data, where crossovers increase in distal regions and decrease within the pericentromeres.

For Col/Ler hybrids our FTL and GBS crossover measurements are in close agreement. The *11b* interval increased crossover frequency by 26.1% and 39.5% in the *msh2* mutant (5.4 to 6.9 cM and 4.8 to 6.7 cM), respectively, and decreased by 8.6% and 4.7% within *5.10* (19.6 to 17.9 cM and 21.0 to 20.1 cM), respectively. For the Col/CLC hybrid, the FTL and GBS data are in close agreement within the *5.10* interval, decreasing by 4.5% and 1.1% in *msh2* (25.4 to 24.3 cM and 25.4 to 25.1 cM), respectively. However, crossover frequency differed within the *11b* interval in the Col/CLC background. Although *11b* crossover frequency increased by 103.5% when measured using FTL reporters (9.0 to 18.4 cM), there was only a slight decrease of 1.05% in the GBS data (3.2 to 3.1 cM). These comparisons indicate that crossover changes in *msh2* are broadly consistent between the different methodologies, although there are notable exceptions.

There are several reasons why GBS and FTL crossover measurements may differ:

- (i) Crossover measurements made using FTL intervals are calculated from many thousands of meioses and provide a high confidence measurement within a specific region. In contrast, GBS samples a lower number of meiosis, but identifies crossovers throughout the entire genome. We note that the *11b* FTL interval is relatively narrow (1.85 Mb), equivalent to ~1.6% of the genome. Hence, GBS measurements are likely to be less reliable at this physical scale compared to FTL measurements, due to the difference in depth of scoring.
- (ii) As discussed earlier, Arabidopsis male and female meiosis show differences in crossover frequency (Giraut et al., 2011), with male meiosis showing high recombination specifically in the distal sub-telomeric regions. Our GBS data are derived from sequencing of F₂ individuals and thus represent an average of male and female meiosis, whereas the *11b* FTL interval only measures male meiosis. Interestingly, Col/Cvi hybrids (Cvi comprises the largest proportion of the CLC genome) were previously observed to show highly elevated crossover frequency within *11b* (Ziolkowski et al., 2015). Hence, the elevated *11b* crossover frequency that we observe compared to GBS in Col/CLC hybrids, could be due to a differential effect during male meiosis compared to female.

To strengthen this aspect of the paper we provide additional data using the centromeric *CEN3* FTL interval, comparing wild type and *msh2* Col/Ct hybrids (Fig. S8 and Table S9). Notably this is a pollen-FTL interval and so measures crossover frequency specifically in male meiosis. Consistent with our GBS-based observations that crossovers decrease within the pericentromeres in *msh2*, we observe that *CEN3* shows a significant decrease in crossover frequency in *msh2* compared to wild type Col/Ct hybrids (t-test $P=3.31\times 10^{-6}$) (Fig. S8 and Table S9). The two methods were broadly in agreement within the *CEN3* interval, as crossover frequency decreased by 56.3% and 27.6% in the *msh2* mutant in the FTL and GBS data (9.9 to 4.3 cM and 10.7 to 7.7 cM), respectively. In contrast to *5.10*, we note that *CEN3* more tightly spans the pericentromere, and does not include significant proportions of the adjacent chromosome arm regions.

To summarise, we observe that *msh2* causes increased crossover frequency within the sub-telomeric *11b* (male meiosis) and *420* (male + female meiosis) intervals, whilst *msh2* decreases crossover frequency within the pericentromeric *CEN3* (male meiosis) and *5.10* (male + female meiosis) intervals, in hybrid backgrounds. Together, these results are consistent with the crossover remodelling that we observe genome-wide in *msh2* using GBS (Fig. 4B-4C). Therefore, we argue that despite the limitations of comparing GBS crossover maps to FTL measurements discussed above, our data using these two approaches are in broad agreement.

6. While unlikely, it is formally possible that DSB distributions are affected by MSH2

genotype. This should be discussed.

We agree with the reviewer that this is a possibility, which we had previously mentioned in our Discussion but did not elaborate on. We have now extended consideration of this model and present a graphical representation in Figure 8B. Supporting this model, work in budding yeast has shown that higher DSB levels occur in *zmm* mutants, where recombination progression is delayed (Thacker et al., 2014). Indeed, feedback processes triggered by mismatches and MSH2 recognition may be more significant in Arabidopsis, given the greater variation in SNP density observed across the chromosomes (Fig. 8D). We discuss the potential for gradients of SNPs along the Arabidopsis chromosomes to trigger DSB feedback mechanisms differentially between regions.

7. In many of the plots, X and Y axis values are in a font that is so small as to be unreadable.

We have increased font sizes throughout the figures to improve legibility.

Referee #2:

This is a very interesting study, of high quality and very exhaustive, on the effect of sequence polymorphism on homologous recombination in plants and on the role of MSH2 in determining the distribution of crossovers as a function of sequence divergence at the local level or at the whole chromosome level. The data presented here are novel and this topic is of general interest for geneticists working on sexually reproducing organisms.

We are pleased that the reviewer finds our work of interest.

This work comes in the background of apparently conflicting evidence in the field on this topic. On the one hand, evidence in the literature of a suppressor effect of sequence divergence on crossovers that is mediated by MSH2 and in this work, evidence for a pro-crossover effect of MSH2. While the work presented here is quite impressive, I feel that the authors should have made a greater effort to discuss if these two clashing views can be reconciled or if one of them is plain wrong or maybe both are wrong - I elaborate on this below (point 5). I also raise a number of points that the authors should consider.

We propose that the biochemical function of MSH2 MutS heterodimers is highly conserved. Specifically, MSH2 heterodimers are able to bind mismatched DNA sequences, which triggers ATP binding and a conformational change (Modrich and Lahue, 1996; Kunkel and Erie, 2005). However, the consequences of MSH2-dependent mismatch recognition are likely to vary between species, cell type, cell cycle stage and according to the structure of the mismatched DNA molecule.

In the Discussion we compare Arabidopsis and budding yeast *msh2* phenotypes and compare their genomes in Figures 8C-8D. For example, these species differ in terms of crossover frequency, SNP distributions, chromosome size and chromatin states. This combination of factors likely contributes to differences in *msh2* meiotic recombination phenotypes. Indeed, different recombination phenotypes between orthologous factors have previously been observed between Arabidopsis and budding yeast. For example, *recq4a recq4b* in Arabidopsis shows a 3.3 fold increase in Class II crossovers and retains wild type fertility levels (Séguéla-Arnaud et al., 2016; Serra et al., 2018b). Whereas budding yeast *sgs1* mutants show a 1.6 fold crossover decrease and aberrant joint molecules accumulate in late prophase I that reduce fertility (Zakharyevich et al., 2012; Oh et al., 2007; Rockmill et al., 2003; De Muyt et al., 2012; Jessop et al., 2006a).

Although chromatin exerts a significant effect on meiotic recombination in both budding yeast and Arabidopsis, the nature of their chromatin landscapes differ greatly (Yelina et al., 2015; Choi et al., 2018; Underwood et al., 2018; Pan et al., 2011; Mimitou et al., 2017; Acquaviva et al., 2013; Wu

and Lichten, 1994). For example, megabase-scale regions of heterochromatin surround the Arabidopsis centromeres, which are strongly modified by DNA methylation and H3K9me2 and are suppressed for meiotic recombination. In contrast, these chromatin marks do not exist in budding yeast, where centromeres consist of a single nucleosome that lacks extensive flanking heterochromatin. However, in other respects the influence of chromatin is similar between these species. For example, sequencing of SPO11-1-oligos in both species has revealed DSB hotspots in the nucleosome-free regions of gene promoters (Choi et al., 2018; Pan et al., 2011; Wu and Lichten, 1994).

A further significant difference between Arabidopsis and budding yeast is the distribution of polymorphisms along the chromosomes. In Figure 8D we compare SNP density per 10 kb along the budding yeast and Arabidopsis genomes. Greater regional variation in SNP density occurs along the larger Arabidopsis chromosomes, which may cause an effect on progression of recombination. For example, in our DSB model (Fig. 8B), high polymorphism density is proposed to inhibit repair of strand invasions via MSH2 heterodimers. This may then cause recruitment of additional DSBs to these regions via feedback mechanisms, ultimately leading to higher crossovers. Hence, gradients of SNP density along the chromosomes may differentially trigger feedback processes that contribute to the observed differences in *msh2* recombination outcomes between species.

1- I am wondering if the TIGER pipeline could underestimate the number of crossovers in less divergent regions? Usually Markov-based algorithms are more effective at determining the transition from one parental type to the other when there is more divergence in sequence. The borders of the CO in such cases might be very large and might have been omitted from some of the analyses.

We acknowledge that SNP density will have an effect on the performance of TIGER in crossover identification. In this regard we refer the reviewer to simulations performed in the original TIGER study (Rowan et al., (2015) G3). For these simulations, a set of known crossovers were pre-defined and it was then assessed how accurately TIGER was able to detect them. At 0.1x sequencing coverage, TIGER recovered 97.5% of the crossovers. At 10x coverage, it recovered 99.3% of the crossovers. Given the typical coverage per sample in the datasets used here (~1-2x), we estimate that we likely miss between 1-2% of crossovers. Hence, even if all of these missing events occurred in the less diverse regions, they would only account for a small percentage of the total crossovers and are unlikely to obscure the patterns we observe.

We also note that the crossover increases observed in *recq4a recq4b* are highest in the least SNP dense regions, indicating that we have adequate ability to detect crossovers despite lower SNP density. We also note that the crossover numbers identified using our genotyping-by-sequencing and TIGER pipeline are in agreement with independent estimates of crossover numbers, based on genetic mapping and cytological estimates (Fernandes et al., 2017; Giraut et al., 2011; Ferdous et al., 2012; Chelysheva et al., 2010).

It is true that crossover intervals in less SNP diverse regions will tend to be wider. So, by excluding events greater than 10 kb we may bias against crossovers in these regions. However, we note that the 10 kb filter we apply leads to relatively small changes in the number of crossovers analysed. For example, the number of crossovers filtered on the basis of >10 kb width were between 0.2 and 5.3% of the total.

ColxLer Serra = 1,840 vs 1,814 crossovers (1.5%)
ColxBur = 1,396 vs 1,393 crossovers (0.2%).
ColxWs = 1,485 vs 1,428 crossovers (3.8%).
ColxCLC = 1,587 vs 1,579 crossovers (0.5%).
ColxCt = 2,478 vs 2,445 crossovers (1.3%).
ColxLer Rowan = 17,077 vs 16,175 crossovers (5.3%).

2- In larger genomes than Arabidopsis there is a very large portion of the chromosome around the centromere with almost no recombination. I am curious if the authors have any thoughts as to why pericentric region close to the centromere have the highest degree of CO, and why we see this only in Arabidopsis.

The reviewer notes that plant species with large genomes, for example wheat, barley, maize and tomato, show extensive regions of crossover suppression surrounding the centromeres (Higgins et al., 2012; Choulet et al., 2014; Demirci et al., 2017; Li et al., 2015). Crossover suppression in these regions correlates with the presence of heterochromatic modifications, including DNA methylation and H3K9me2 (Higgins et al., 2012; Choulet et al., 2014; Demirci et al., 2017; Li et al., 2015). As noted earlier, the centromeric regions in Arabidopsis are also heterochromatic and show high levels of DNA methylation, H3K9me2 and nucleosome occupancy, as well as suppressed DSBs and crossovers (Underwood et al., 2018; Choi et al., 2018; Yelina et al., 2012, 2015; Lambing et al., 2020). However, the physical extent of pericentromeric heterochromatin in Arabidopsis is small relative to that of euchromatin, compared to plant species with large genomes. It is also noteworthy that most plant species, despite showing extensive variation in physical genome size, typically experience ~1-2 crossovers per chromosome (Mercier et al., 2015). As a consequence, the Arabidopsis genome shows relatively high crossover frequency (~4-5 cM/Mb) compared to plants with large genomes (e.g. wheat ~0.1-0.2 cM/Mb) (Choulet et al., 2014; Serra et al., 2018b). We propose that differences in relative euchromatin and heterochromatin content between these species contributes to differences in their crossover landscapes, along the telomere-centromere axes.

To illustrate relationships between recombination, polymorphism and chromatin, we have added a new Figure 2 that compares crossover and SNP frequency along the Arabidopsis genome with other genome-wide datasets, including SPO11-1-oligos, DNA methylation, H3K9me2, H3K4me3, nucleosomes (MNase-seq) and gene density (Choi et al., 2018, 2016; Lambing et al., 2020; Yelina et al., 2015). This shows that despite regions close to the centromeres containing substantial sequence polymorphism, the effect of chromatin dominantly suppresses recombination. We also note that our new GLM modeling (discussed in response to reviewer 1) shows a parabolic relationship between SNP density and crossovers (see main Fig. 2). In this model increasing SNP density associates positively with crossovers, until higher SNPs/kb values are reached when the relationship becomes negative.

Could this reflect pairing patterns starting at centromeres? That these regions are more polymorphic is the explanations of the authors, but then why this is not the case in other species. Please discuss that.

The Arabidopsis centromeres have been observed to undergo pairing during prophase (Armstrong et al., 2001; Da Ines et al., 2012). Specifically, clustering of centromeric heterochromatin occurs during zygotene which persists until pachytene (Armstrong et al., 2001; Da Ines et al., 2012). Hence, these centromere pairing interactions could also promote recombination in the Arabidopsis pericentromeres. We have added a note on this point to the manuscript.

3- The authors extend previous works from their and other labs by showing that in all crosses COs are more frequent in promoters and terminators which are also more polymorphic than exons. Could it be the explanation for this local effect of recombination in the diverse region (which would have little to do with polymorphism itself but be an indirect effect of chromatin accessibility). What is shown locally at the bp level might be quite different from the phenomenon seen at the whole chromosome level.

This analysis was included in the crossover-associated motif section, which in response to reviewer 1 has now been removed. We acknowledge the reviewer's points concerning the importance of chromatin for recombination. As discussed earlier, we have now added plots of chromatin, crossovers and SNPs to Figure S2 to illustrate these patterns, at the chromosome

scale.

4- Any insight on the effect of *msh2* on the reduction of CO in pericentric regions (Fig 3)?

We present two models to explain the observed remodelling of crossovers in *msh2* in Figure 8A and 8B. The first model proposes that MSH2-mediated recognition of mismatched strand invasion events promotes designation of Class I ZMM crossover repair. The second model proposes that MSH2 recognition of mismatches in joint molecules promotes disassembly or slows progression of repair and causes feedback signaling to DSBs. Feedback would then increase DSBs locally and thereby increase crossovers. We have further considered these models and supporting evidence in the Discussion.

5- Figure 6 is an analysis of the effect of sequence divergence and MSH2 at the chromosomal scale. Both the effect of divergence in one chromosomal segment on CO and the effect of MSH2 are dependent on the rest of the chromosome. Remarkably the effect of *msh2* on CO in the subtelomeric region of chromosome 3 is inverse depending on divergence in the rest of the genome (e.g. Fig 6C HET-HOM versus HOM-HET). This is a very interesting finding but here I am really wondering if one can think of a mechanism that would explain both the local effect and the chromosomal effect-seems like two different phenomena? Please elaborate on that. It seems to me that the authors have not provided any satisfactory mechanistic model that would explain both phenomena.

We are pleased that the reviewer finds these observations of interest. Our previous work identified that juxtaposition of heterozygous with homozygous regions, at the megabase scale, causes crossover increases in the heterozygous region, at the expense of the adjacent homozygous region (Ziolkowski et al., 2015). This effect was shown to be mediated via the Class I repair pathway (Ziolkowski et al., 2015). The juxtaposition effect represents a context where heterozygosity leads to relative promotion of crossover repair. As MSH2 heterodimers mediate mismatch recognition, this was the major impetus for us to investigate the effect of *msh2* on this phenomenon. Indeed, by repeating the juxtaposition experiment in *msh2*, we show that heterozygosity no longer causes increased crossover frequency when juxtaposed with adjacent homozygous regions. We have modified the introduction to the juxtaposition experiment in the results section to explain the rationale for testing *msh2* in this assay more clearly.

From mapping crossovers via sequencing, we show that in five independent wild type populations a positive correlation exists between crossover frequency and SNP density, at the chromosome scale. We show that this correlation is weakened in *msh2*. We cannot currently prove that these sequencing-based observations are mechanistically linked to the juxtaposition effect experiment, but we propose they are connected. This proposition is based on the fact that both phenomena occur at the megabases-scale and that *msh2* and *HEI10-OE* cause similar effects on the relationship between crossovers and polymorphism in both contexts.

I also think that the title is not reflecting the authors results. I would say that "the effect of MSH2 on CO in one given region is dependent on the sequence divergence surrounding this region" is a more appropriate description of the authors findings.

We appreciate the reviewer's thoughts and their suggested title. However, we think the juxtaposition aspect is only clearly shown by the data in Figure 7 and so does not warrant refocusing the title solely on these results. However, the title has been modified in response to other comments and is now 'MSH2 promotes meiotic crossover formation in regions of high sequence diversity in Arabidopsis'.

Presenting things that way could explain discrepancies in conclusions with other works on the same subject: I note that the results with the 420 FTL are similar to what was reported by Emmanuel et al. 2006 who showed an increase in meiotic CO in *msh2* in the HET-HET situation compared to HOM-HOM and reached the conclusion that MSH2 has an

anti-recombination effect when looking at a specific interval in a specific genetic background configuration.

We entirely agree with the reviewer on this point. The analysis in Emmanuel et al (2006) is consistent with the data we present in this manuscript. In Emmanuel et al (2006) the authors measured crossover frequency using an FTL *Le5-11/22*, which is located in the sub-telomere of chromosome 5 (2.05-4.18 megabases). *Le5-11/22* was analysed in BC₁F₂ populations that were wild type or *msh2*, and Col/Ler heterozygous within the FTL interval. A significant increase in *Le5-11/22* cM from 6.31 to 8.81 was observed in *msh2*. As noted by the reviewer this is consistent with our FTL analysis (Fig. 7C and S8), and genome-wide mapping of crossovers (Fig. 4), which show increased crossover frequency in distal chromosome regions in *msh2*. We now cite the Emmanuel et al paper in the Introduction and Discussion.

I think this is the main story of this work, that the MSH2 effect can be either pro or anti recombination depending on the flanking region heterozygosity. This is what the authors show us and this might settle a long-held controversy. For some reason they decided to have a simple bottom line that ignores their own results.

We agree with the reviewer that these results are important and to emphasize them we have added additional material to introduce this results section and clarified why *msh2* was tested in this assay.

At the gene-scale I have the feeling that what we see is simply due to the strong pro-recombination effect of promoters which are nucleosome depleted, AT rich etc.. and happen to be more polymorphic than exons. At the chromosomal scale we have a much more puzzling story. I am not sure if the authors have enough data but if they could find gene-free regions that undergo CO, they could use it as a control to show that this is polymorphism per se and not gene features that determine CO position. They could be in for some surprises and find that polymorphism has an anti-recombination effect when in non-gene regions.

We agree with the reviewer that this is an interesting question, however we think our crossover data are not sufficiently deep to meaningfully perform this type of sub-setting analysis.

To summarize, I find this work to be very interesting and I laud the authors for tackling a very difficult topic. I do not ask for new data, but for a reexamination of their results and of their interpretation as described above.

We thank the reviewer for their positive comments. We hope that our revised manuscript has addressed the concerns they raised.

Referee #3:

In this paper Blackwell and co-workers investigate the relationship between sequence polymorphism and meiotic crossover frequency in *Arabidopsis thaliana* and they investigate the role of the mismatch repair protein MSH2 in meiotic recombination.

The authors first studied the correlation between CO frequency and SNP density in a number of genetic contexts (four different accessions and several mutant backgrounds) by mapping COs after DNA sequencing of F2 populations. Then they studied the effect of the MSH2 protein (a demonstrated component of the mismatch repair machinery) on meiotic recombination patterns and provide information on the dynamics of this protein during meiosis, based on immunofluorescence studies.

This paper represents a very nice piece of work, with lot of interesting and globally robust data; data are clearly presented and the paper is extremely well written. While I agree with

many of the provided conclusions, I question the ones that concern the link between sequence polymorphisms and CO levels. These are too often overstated, starting with the title of the paper itself.

We thank the reviewer for their positive comments. We believe we have clearly shown that the crossover landscape is remodelled in *msh2*, and that these changes relate to underlying patterns of SNP density. Specifically, we report increased *msh2* crossovers in lower diversity sub-telomeric and interstitial regions at the expense of the higher diversity pericentromeric regions. However, we acknowledge that the causes of these observations remain unclear. As explained below and in response to the other reviewers, we now more fully consider the joint effects of SNP density, chromatin and telomere/centromere pairing on the crossover landscape in wild type and *msh2*.

We have modified the title of the paper as follows: 'MSH2 promotes meiotic crossover formation in regions of high sequence diversity in Arabidopsis'.

My second concern, is that most of the data analyzed here have been generated based on F2 population sequencing. In consequence, recombination rates and profiles analysed correspond to an addition of male and female recombination. This is a problem for two things. First, it is well known that male and female meiosis have very contrasted patterns of recombination both in terms of rates and patterns (Giraut et al 2011). Second, it is also known that this sexual dimorphism is abolished in mutants such as *recq4ab* (Fernandes et al 2018), a genetic context used here to demonstrate that Class II COs are polymorphism insensitive.

We agree with the reviewer that there is the potential for sex-specific *msh2* recombination phenotypes. We show in interval *11b* (a distal FTL interval measuring male meiosis) that crossover frequency increases in *msh2* hybrids compared to wild type. This is consistent with crossover remodelling we observe from sequencing F₂ plants, which represents both male and female meiosis (Fig. 4). Furthermore, we have added new *CEN3* FTL data that measures crossover frequency across the pericentromere of chromosome 3 in male meiosis (Fig. S8 and Table S9). *CEN3* crossover frequency showed a decrease in *msh2* compared to wild type in Col/Ct hybrids, which is consistent with the changes we observed from sequencing-based crossover mapping and *5.10* FTL measurements (both of which measure male and female meiosis). Finally, we note that crossover frequency increases in the *msh2* mutant within the distal *420* interval (which measures an average of male and female meiosis) in the Col/Ct background. Together these data are consistent with remodelling of crossovers in *msh2* occurring in both male and female meiosis. However, we acknowledge that we do not have direct evidence for an effect of *msh2* on female meiosis, and it remains a possibility that there are sex-specific differences.

Therefore, I am afraid that many of the correlation data could be considerably biased in a sense that is difficult to estimate. In consequence I suggest that the authors keep the extrapolation of their results to establish a link between CO rates and SNP density to the discussion only.

We have extensively rewritten the manuscript to moderate interpretation of our data.

More detailed comments

- Studies at global (200 kb) scale: The authors do demonstrate that there is a correlation between the degree of polymorphisms and the rate of COs but they shouldn't hide/minimize the fact that the correlation is extremely weak. Besides, even if CO frequency is the highest in pericentromeric regions where SNP density is the strongest, it is not true in subtelomeric regions (lowest SNP density for high CO density). Therefore, I think that the conclusion of this paragraph (1196-197) is misleading and does not take all the data into account. Even if I agree that the authors revealed a correlation between

crossover rates and high SNP density, the correlation could be easily biased by the fact that the highest SNP density is found in pericentromeric regions which are well known to be different from the other parts of the chromosomes for many other things than SNP densities (recombination dynamics, DNA and chromatin marks, repeated element composition etc...). If high CO rates in these regions is driven by any of these parameter (and not polymorphisms), even partially, it will introduce a bias toward the strongest polymorphic regions. This should be considered to analyse the results, to draw conclusions and in the discussion.

The reviewer raises a number of important points concerning co-variation of multiple factors along the telomere-centromere axes of the Arabidopsis chromosomes, including chromatin, polymorphism and crossovers. We fully acknowledge that chromatin plays an important role controlling recombination, and is necessary to consider jointly with polymorphism. For example, we have previously demonstrated the role of chromatin in shaping the Arabidopsis recombination landscape (Choi et al., 2018; Underwood et al., 2018; Lambing et al., 2020; Yelina et al., 2015). The Arabidopsis centromeric regions are heterochromatic and show high levels of DNA methylation, H3K9me2 and nucleosome occupancy, as well as suppressed DSBs and crossovers. Hence, despite these regions containing substantial polymorphism, the effect of chromatin will dominantly suppress recombination. To illustrate these correlations we have added a new Figure 2 that compares Col/Ler crossover and SNP frequency along the genome with several additional genome-wide datasets, including SPO11-1-oligos, nucleosomes (MNase-seq), H3K9me2, DNA methylation, H3K4me3 and gene density (Choi et al., 2018, 2016; Lambing et al., 2020; Yelina et al., 2015). We have discussed these relationships in the main text and Discussion.

As explained in response to reviewer 1, to quantitatively model the effects of multiple parameters on crossovers, including SNP density and chromatin, we used a generalized linear model (GLM), also known as a multivariate regression model. This is an extension of a previous model where we examined the relationship between crossovers, meiotic DSBs (SPO11-1-oligos) and chromatin (Choi et al., 2018 *Genome Res*). In this model we consider all SNP intervals where it is possible to detect a crossover, corresponding to 534,780 intervals with a mean width of 224 bp. The binary response variable in the model is whether at least one crossover was observed in a given SNP interval. We then calculate explanatory variables for the same intervals, including SPO11-1-oligonucleotides, nucleosomes (MNase-seq), H3K4me3, DNA methylation and SNP density. For SNP density we calculated a rolling average of SNPs/kb with a 1 bp step and used these values to calculate mean SNPs/kb per interval. Data were modeled with the `glm2` function in R, using the binomial family with the logit link function, using the following formula:

$$\text{Crossovers} \sim (\text{SPO11-1} + \text{nucleosomes} + \text{H3K4me3} + \text{DNA methylation} + \text{SNPs/kb} + \text{width})^2$$

For model selection we used the `stepAIC` function from the MASS package in R using both forward and backward directions in order to minimize the AIC. As reported previously, our model shows negative effects for nucleosomes and DNA methylation on crossovers, and positive effects for SPO11-1-oligos (Fig. 2B). Interestingly, we observe that SNPs/kb shows a parabolic relationship with crossovers. Initially, there is a positive relationship with increasing SNPs/kb showing a positive effect on crossovers. However, beyond a certain threshold, the relationship becomes negative. We interpret this as SNP density having a positive effect on the chance of crossover, up until a threshold when high polymorphism becomes inhibitory. We note that this is consistent with the suppressive effect of larger structural polymorphism on crossovers (Rowan et al., 2019). We also previously observed that within the *RAC1* and *RPP13* resistance genes, high local SNP density associates with suppressed crossover at the fine scale (Serra et al., 2018a). We have added results from our GLM analysis to Figure 2B and Table S5. We also present updated correlation analysis of crossover frequency and polymorphisms to Figure 2A, which further supports a parabolic relationship between SNPs and recombination.

In response to the reviewer's point, we would also like to highlight our heterozygosity juxtaposition experiments (Fig. 7). Here we varied the presence or absence of polymorphism

within a distal region of chromosome 3 (420). Therefore, this experiment avoids many of the possible confounding variables present in genome-wide analyses. In this assay the *msh2* mutation prevented increases in crossover frequency caused by juxtaposition of heterozygosity occurring at the megabase scale.

-Studies at local scale (a few kbs)

Here the authors analyzed SNP density in windows of increasing sizes surrounding the detected COs. They detected a very clear enrichment in SNPs around COs. However, I think it would be interesting to analyze separately COs that occur in pericentromeres from the other ones. If the enrichment in SNP is not true for COs outside the pericentromeric regions it means that the correlation observed between SNP density and CO levels is biased by the elevated rates of recombination in pericentromeres which happens to be SNP rich. If it is the case, the conclusion of the paragraph (l214-215) as well as the general conclusions of the manuscript should be modified (see also my remark above).

We have repeated analysis of SNP density around crossovers separately for those located in the chromosome arms and pericentromeres, as requested. These analyses are provided in Figure S3. SNP enrichment around the crossovers is observed compared to random positions in the crossovers from the chromosome arms or the pericentromeres, in the large majority of cases. As expected, SNPs/kb values are on average higher in the more diverse pericentromeric crossovers, compared to the chromosome arms. We note that *msh2* reduces SNP density around crossover sites in both the chromosome arm and pericentromere contexts, for all three hybrid backgrounds. Although the Col/Ler (Serra) crossovers in the chromosome arms were only significantly different from random for interval widths <2 kb.

- Then the authors perform the same kind of analyses in genetic backgrounds showing an increase in either Class I (HEI10 overexpressor) or Class II (*recq4ab*) or both (HEI10 *recq4ab*) mutants. While high SNP density is associated with COs in wt and HEI10 background, it is not the case anymore neither for *recq4ab* nor for HEI10 *recq4ab* backgrounds. the authors conclude that these data are consistent with class I COs that associate with regions of strong polymorphisms but they do not explain why no difference is observed between the *recq4ab* and HEI10 *recq4ab* genotypes.

Previously we did not explicitly test for significant differences between SNPs/kb values observed around crossovers in *recq4a recq4b* compared with *HEI10-OE recq4a recq4b*. This comparison has now been performed and a significant difference between these genotypes observed (right hand section of Fig. 3C). Specifically, *HEI10-OE recq4a recq4b* shows higher SNPs/kb values around crossovers than *recq4a recq4b*, which supports our conclusions.

In addition, how do they explain the shape of the curve (fig2C) in *recq4ab* (and HEI10 *recq4ab*) that deviates from the random situation for large window sizes?

The reviewer notes that for windows in the size range ~4-10 kb, SNPs/kb values for crossovers in *HEI10-OE recq4a recq4b* and *recq4a recq4b* decrease below the random expectation. In *recq4a recq4b* backgrounds there is a large increase in Class II crossovers, which we propose have a preference to form in the distal regions with low SNP density. Notably however, Class I crossovers still form in both of these genetic backgrounds, which may contribute to the rise in SNPs/kb in *HEI10-OE recq4a recq4b* and *recq4a recq4b* as crossover midpoints are approached.

Once again it would be interesting to test separately COs in pericentromeric regions from the others to see if both show an enrichment in SNP density only when Class I COs predominate.

We have repeated this analysis for all genotypes after separating crossovers into chromosome arms and pericentromeres (Fig. S3). In the backgrounds where Class I crossovers predominate

(wild type and *HEI10-OE*) we observe a significant enrichment of SNPs/kb values around crossovers compared to random, for the majority of windows tested.

And here too the question of ignoring the different recombination profiles of male and female meiosis is a big issue, notably considering that it has been shown that *recQ4ab* mutations impact differently male and female meiosis (Fernandes et al 2018).

Arabidopsis male meiosis has additional crossovers compared to female (map length is 575 cM versus 332 cM), with the additional recombination events occurring predominantly in the sub-telomeric regions (Giraut et al., 2011). Recent work has indicated that this sex difference is dependent on the Class I crossover pathway (Fernandes et al., 2017). As explained earlier, we have now added new FTL data using pollen-based reporters, which allow us to specifically assess the *msh2* phenotype in male meiosis. We observe that crossover frequency within the centromeric interval *CEN3* decreases in *msh2* Col/Ct hybrids compared to wild type. Conversely, we observe that crossover frequency is increased within the male-specific sub-telomeric *11b* reporter. Therefore, we demonstrate a clear effect of *msh2* during male meiosis, where crossovers become distalized and remodel away from the relatively diverse pericentromeres. As we also observe the same relationships and patterns in our data that represents the average of male and female meiosis (i.e. GBS crossover mapping and the *5.10* and *420* reporter intervals), we argue that this is consistent with remodeling of the *msh2* crossover landscape occurring during both male and female meiosis. However, we acknowledge that we do not have direct evidence to demonstrate the effect of *msh2* on female meiosis, and it remains a possibility that there are sex-specific differences.

- *msh2* mutant analyses

The change in recombination profiles observed in the *msh2* F2 populations is very convincing, with a tendency to relocate COs from pericentromeric to the arms. However, I didn't find the conclusions drawn out of the analyses at a local scale convincing. The authors observed a reduced SNP enrichment at CO sites in *msh2* only for the Col/Ler population. For the two others, the decrease is very mild (Col/Ct and Col/CLC). therefore, the conclusion drawn 1338 and 339 is very surprising: "This is consistent with *msh2* crossovers forming in regions of elevated meiotic DSBs, but with a reduced association with higher SNP density relative to wild type." This does not fit with the data.

We agree with the reviewer that the crossover remodeling we observe at the chromosome scale is very consistent across the three populations analysed (Col/Ler, Col/Ct and Col/CLC). We also acknowledge that at the fine-scale the effect of *msh2* on SNPs/kb enrichment around crossovers is more variable. This indicates that some aspects of the *msh2* recombination phenotype are modified by genetic background. However, in each case SNPs/kb enrichment is significantly reduced in *msh2*, albeit to varying degrees. This statistical analysis is now presented in the lower part of Fig. 5B. Here SNPs/kb values around crossovers were significantly different between wild type and *msh2*, for all expect the first window in the Col/Ct and Col/CLC populations.

-The FTL data are a bit confusing which is a pity because they should be a way to analyze male recombination independently from female.

We apologise for this confusion. As discussed above, we now provide FTL data using the pericentromeric interval *CEN3* in Col/Ct hybrids. Importantly, *CEN3* is a pollen-based FTL, so provides a male specific estimate of crossover frequency. We show that *CEN3* undergoes a significant decrease in crossover frequency in *msh2* compared to wild type, in Col/Ct hybrid backgrounds (Fig. S8 and Table S9). This is further consistent with crossovers remodelling away from the pericentromeres to the chromosome arms in *msh2*. As we observe consistent patterns of crossover change in *msh2* using *11b* and *CEN3* (male meiosis), and *5.10*, *420* and GBS mapping (male and female), we propose that male and female meiotic recombination landscapes are both likely to be altered in *msh2*. However, whilst we provide clear evidence for *msh2* crossover remodelling during male meiosis, we acknowledge that we have not formally demonstrated the

msh2 crossover remodelling phenotype during female meiosis.

Why did the authors choose the 510 interval that indeed encompass Chr5 centromere but also largely extends over chromosome arm?

At the time of crossing, 5.10 was the most suitable FTL interval that we had available that spanned a centromere. As discussed above, we have now added new FTL data using the pericentromeric FTL interval *CEN3* in Col/Ct hybrids. We show that *CEN3* undergoes a significant decrease in crossover frequency in *msh2* Col/Ct compared to wild type Col/Ct (Fig. S8 and Table S9). Importantly, *CEN3* more tightly spans the pericentromere in comparison with 5.10 (Fig. S8), making this experiment a better demonstration of this point.

Here it is clear that a better description of the regions denominated as "pericentromeres" should be provided (and not only refer to Choi et al).

In previous work we defined the centromeres as the contiguous regions flanking the TAIR10 assembly gaps that show an absence of crossovers in wild type (Copenhaver et al., 1999; Giraut et al., 2011; Salomé et al., 2012). We define the pericentromeres as the contiguous regions that surround the centromere assembly gap with higher than average DNA methylation. The euchromatic arms constitute the remainder of the chromosomes, from the telomeres to the boundaries of the pericentromeres. These coordinates were previously published in Table S9 from Underwood et al., (2018), which we reproduce below. In response to reviewer 1 we have provided a new Figure S2 comparing chromosome profiles of crossover and SNP frequency in relation to chromatin datasets. The plots in Figure S2 are marked to indicate the pericentromere boundaries, as listed in the table below.

Chr	North Arm	North Pericentromere	Centromere	South Pericentromere	South Arm
1	1 - 11,420,000	11,420,001 - 13,920,000	13,920,001- 15,970,000	15,970,001 - 18,270,000	18,270,001 – 30,427,671
2	1 – 910,000	910,001 – 2,950,000	2,950,001 - 4,750,000	4,750,001 - 7,320,000	7,320,001 – 19,698,289
3	1 – 10,390,000	10,390,001 12,680,000	12,680,001 – 14,750,000	14,750,001 - 16,730,000	16,730,001 – 23,459,830
4	1 – 1,070,000	1,070,001 – 3,390,000	3,390,001 – 4,820,000	4,820,001 – 6,630,000	6,630,001 – 18,585,056
5	1 – 8,890,000	8,890,001 – 10,950,000	10,950,001 – 13,240,000	13,240,001 – 15,550,000	15,550,001 – 26,975,502

As for the *11b* interval, the huge increase observed in Col/CLC background is difficult to understand. The male/female interpretation is poorly convincing because it is expected to act on both wild type and *msh2*. Here too the conclusion of the paragraph is not correct and largely overstated (I308/309).

We agree with the reviewer that the ~2 fold cM increase observed in *msh2* within *11b* in Col/CLC hybrids is larger than estimated from our GBS based crossover measurements. The CLC background is predominantly Cvi and we previously reported that the FTL intervals *11b*, *12f*, and *CEN3* are significantly elevated in Col/Cvi hybrids, compared to inbreds and other hybrids (Ziolkowski et al., 2015). As our GBS data sequenced F₂ individuals, it represents crossover data from male and female meiosis. Hence, it is possible that crossovers are lower in *11b* in female meiosis. Additionally, FTL measurements allow 1,000s of meioses to be scored for crossovers within defined intervals from individual plants. In contrast, our GBS approach maps crossovers genome-wide from ~384 meioses per population. Hence, the GBS data may be underpowered in terms of comparing crossover frequency within the relatively narrow physical scale of *11b*.

- The MSH2 immunofluorescence study is a bit superficial. From what I can see from figure 5, MSH2 is more detected as a thread-like signal than as foci.

We agree with the reviewer that MSH2 antibody signal is not linear to the degree observed for ASY1. However, the MSH2 signal clearly tracks along the chromosome axes, as well as localizing to higher abundance foci. Interestingly, HEI10 has been reported to show a similar staining pattern during Arabidopsis meiosis (Hurel et al., (2018) *The Plant Journal*).

In consequence, it is not surprising that MSH4 foci mostly colocalize with the MSH2 signal. If authors are convinced that MSH2 form foci, then they should provide better pictures and quantification as well as proper colocalization studies (counting, comparison to random colocalisation etc...).

The reviewer is correct that due to the abundant signal of MSH2 and MSH4 in meiotic nuclei, there is a high chance of foci co-localization by chance. One method to assess overlap is to randomize the data via rotation of one of the images, whilst leaving the other image in place, and then re-counting foci that co-localise. We had hoped to complete this additional analysis before resubmission. However, due to the ongoing COVID-19 pandemic the laboratory of James Higgins at the University of Leicester remains under lockdown. Due to James and his team not being able to access the necessary computers and software to perform this analysis we have not yet been able to do this. If the reviewer feels this additional analysis is essential for publication, then we hope to be able to provide this analysis as soon as James' lab reopens.

Beside a correct staging of meiocytes for which MSH2 signal is detected should be provided (using a marker that allows unambiguous staging of the meiocytes such as ZYP1).

In this analysis we staged meiocytes using DAPI-stained chromosome morphology and criteria including the thickness of the chromosomes (synapsed or unsynapsed), the position of the nucleolus and the shape of nucleus.

Minor comments

- Abstract and title should be revisited except if the authors provide additional analyses that show a clear causal relationship between sequence polymorphism and CO rates.

We have modified the title to: 'MSH2 promotes meiotic crossover formation in regions of high sequence diversity in Arabidopsis'. We feel this updated title is consistent with our data and describes the fundamental observations that we report in the paper. We provide two alternative models in the Discussion, Figure 8A and 8B, which will serve as hypotheses for future investigations.

-I wouldn't use "HEI10" for the HEI10 overexpressor line (it can be confused with the HEI10 gene)

We have followed the reviewer's suggestion and now use *HEI10-OE* (*HEI10* overexpression) throughout the manuscript. We note that we have used *HEI10* to denote overexpression in previous publications (Serra et al 2018, Ziolkowski et al 2017). Therefore, we have added a note to the Materials and Methods section to explain the name change.

- Previous published data on *A.thaliana* MSH2 should be better acknowledged, in the introduction as in the discussion.

We apologise for not providing a more complete representation of published work investigating the role of *MSH2* in Arabidopsis in our manuscript. We have now added additional references to the following work in the Introduction and Discussion.

We include reference to *in vitro* studies testing the binding preferences of MSH2 in heterodimeric complexes with MSH3, MSH6 and MSH7. These MSH2 heterodimers show varying affinity for

base mismatches and short (1-3 bp) indel polymorphisms (Culligan and Hays, 2000; Wu et al., 2003; Adé et al., 2001). Mutation of *msh2* in Arabidopsis has been shown to cause increased mutation rates of single bases and short indels, resulting in deleterious phenotypes following inbreeding (Leonard et al., 2003; Hoffman et al., 2004; Watson et al., 2016; Belfield et al., 2018).

Somatic homologous recombination and its sensitivity to polymorphism has been measured in plants using split *GUS* (*GU:US*) transgenes, with increasing numbers of mismatches introduced between the substrate repeats (Emmanuel et al., 2006). These assays revealed that increasing *GU:US* mismatches decreased HR frequency in wild type, with higher recombination rates observed in *msh2* at a given divergence level (Emmanuel et al., 2006; Li et al., 2006). In budding yeast and mammals similar suppressive effects of mismatches are observed in mitotic HR assays, which are suppressed by *msh2* (Chen and Jinks-Robertson, 1999; Elliott and Jasin, 2001; Datta et al., 1997).

Previous work measured crossover frequency within a 2.1 Mb sub-telomeric *Le5-11/22* FTL interval on chromosome 5 in a Col/Ler BC₁F₂ and observed crossover increases in *msh2* compared to wild type (Emmanuel et al., 2006). This is consistent with our observations showing crossover frequency increases in *msh2*, in the distal sub-telomeres.

- Explain better the CLC background

We now explain CLC in more detail when it is first introduced in the results section. The 'CLC' background is a mosaic of Cvi, Ler and Col accessions, whose genome is comprised predominantly of Cvi sequence, but with a substituted Ler chromosome 5 and additional regions of Ler and Col introgressions on the other chromosomes. These regions are indicated in Figure 1B. The CLC background was under study in our laboratory and provided an interesting genetic background for investigating the effects of mosaic patterns of polymorphism.

- A better description of the CrispCas9 *msh2* allele (*msh2-3* in Ct) should be provided.

We have provided further detail in the main text on how we used CRISPR-Cas9 to generate new *msh2* alleles as follows. We used CRISPR/Cas9 mutagenesis to generate *msh2* alleles *de novo* in the Col and Ct accessions, and Col/Ct recombinant lines (Fig. S5). A pair of gRNAs targeted *MSH2* exon four were designed and introduced downstream of the *U3* and *U6* promoters. These constructs were transformed together with an *ICU2::Cas9* transgene. Transformed T₁ plants were genotyped by PCR amplification with primers flanking the *MSH2* gRNA target sites and sequencing was performed to detect deletions. *msh2* mutants with heritable deletions causing frameshifts, and not carrying the CRISPR-Cas9 transgenes, was identified for further experiments (Fig. S5).

- A clear definition of the pericentromeres is given I186 and reference is made to Choi et al. But it would be important to give the detailed coordinates on the chromosomes of the regions called pericentromeres and show their position on all the figures.

We addressed this point earlier.

- Figure S3 left part is not clear: what is the correlation looked at?

Thank you for identifying this labeling error. Due to a request for modifying how we present these data, the previous Figure S3 has now been removed, and comparison of crossovers with the Poisson expectation is now shown in Figure 1A.

-Table S8: did the authors checked for correct segregations of their markers? Some crosses appear to show a very strong bias among reciprocal phenotypic classes

The reviewer refers to our *11b* FTL flow cytometry data. The reviewer is correct that the

Mendelian expectation for these data is that the two recombinant classes (red-alone and yellow-alone) and the two parental classes (red+yellow and non-colour), should show approximately equal counts, as they represent reciprocal outcomes of meiosis. In this respect we refer to our published work (Yelina et al., 2012, 2013; Ziolkowski et al., 2015), where we show that two aspects of the data make the non-colour and red-alone classes unreliable. First, an excess of non-colour counts are typically observed relative to red+green, which is due to pollen grains that have lost fluorescence from the other classes, for instance due to transgene silencing or damage/inviability. Equally, the red-alone class typically shows an excess of counts relative to yellow-alone, due to 'bleed-over' from the yellow fluorescence. For these reasons we calculate cM values within *11b* using the formula: $cM = 100 \times (N_Y / (N_Y + N_{R+Y}))$, where N_Y is the number of yellow alone pollen, and N_{R+Y} is the number of red and yellow pollen. As a precaution, before performing flow cytometry analysis, all individuals are manually phenotyped under a dissecting fluorescence microscope to confirm transgene hemizyosity and normal expression levels.

- The discussion should provide a deeper examination of several points 1) divergence between male and female recombination rates 2) high recomb rates in subtelomeric regions of the chromosomes 3) consider other possibilities than a causal effect of SNP polymorphisms on preferential CO location

We have addressed these additional points in the Discussion.

- 1) The known differences in male and female crossover landscapes in Arabidopsis are discussed. Arabidopsis male meiosis shows on average 4.5 additional crossovers compared to female, with the additional recombination occurring predominantly in the sub-telomeric regions (Drouaud et al., 2007; Giraut et al., 2011). Recent work has indicated that this sex difference in recombination is dependent on the Class I crossover pathway (Fernandes et al., 2017).
- 2) We refer to the role of telomere pairing and bouquet formation in meiotic recombination. Nucleolus-associated telomere pairing has been observed in early prophase I in Arabidopsis (Armstrong et al., 2001). Higher crossover rates observed in the sub-telomeres may be connected to these observations.
- 3) We fully agree with the reviewer that other genome features exert a significant effect on crossover placement, in addition to SNPs. For example, we have shown that chromatin is a major influence on meiotic recombination. To illustrate this point we provide a new Figure S2 that compares crossovers, SNPs and chromatin at the chromosome scale. We also acknowledge that other factors including crossover interference play a major role in shaping the recombination landscape.

Editorial Advisor's comments (excerpt):

... What could further be considered is that MSH2 in plants has already been characterised to form heterodimers (in vitro) with MSH3, MSH6 and MSH7 (a plant specific MSH6 prologue) (Culligan et al 2000). This is interesting, because the Alani lab showed in 2014 (Rogacheva et al.) that yeast MSH2 / MSH3 heterodimers can stimulate the nuclease activity of the ZMM protein complex MLH1/MLH3 in vitro and thereby act as a pro-CO factor in theory. All this is not discussed in the paper but would certainly substantiate the claims of the authors. A model would place MSH2 (plus a partner protein like MSH3) at mismatched bases in meiotic recombination intermediates recruiting the MLH1/3 heterodimer to promote CO formation. ...

We thank the Editorial Advisor for raising these important points. We have added additional material to the Introduction and Discussion, including the Culligan reference, noting the potential for differential functions between the plant MSH2 sub-complexes and meiotic recombination. Specifically, we discuss the potential that MSH2 heterodimers with MSH3, MSH6 and MSH7,

which have been shown to have distinct *in vitro* mismatch binding preferences, may have different effects on crossover formation at different scales.

Thank you for directing us to the Rogacheva et al study from the Alani group. In the Discussion and Figure 7A we present alternative models that may explain our observations on the *msh2* crossover landscape. The first model is that MSH2 complexes, on recognition of mismatches in joint molecules, can promote the activity of the ZMM pathway. The Rogacheva study demonstrates that purified Mlh1-Mlh3 complexes can act as a metal-dependent endonuclease that is stimulated by Msh2-Msh3. As the MLH1-MLH3 heterodimer is considered part of the ZMM pathway, these results support this model, which we have discussed in the manuscript.

Thank you for submitting your revised manuscript to The EMBO Journal. All three original reviewers have now looked at it again, and found their key criticisms generally satisfactorily addressed. They still retain several minor/specific concerns, as you will see from the comments below, which I would ask you to respond to and/or incorporate during a final round of revision. In particular, referee 2 still finds the title/conclusions somewhat too categorically stated, so please consider whether there may be good ways to reconcile this and rephrase in a more qualified manner. Furthermore, as referee 3 is still concerned about the MSH2 immunofluorescence analyses, it should be helpful to include some additional image analysis (as discussed on page 18 of your response letter) if you would like to keep these data in the (main) manuscript. Regarding referee 3's point about methylation data comparison with somatic vs meiotic tissues, I understand that this concern had not been raised initially, so would not insist on including this - unless of course if it should be straightforward to add. In any case, please again add a brief response letter to the re-reviews upon final resubmission.

During the final revision round, I would kindly ask you to also take care of the following editorial points.

REFEREE REPORTS

Referee #1:

Blackwell and co-authors have addressed my concerns, and I think that their paper is ready to publish. I have the following comments, which can be addressed without a need for re-review.

Line 83-85. Actually, Borts and Haber looked at an artificial hotspot containing pBR322 and URA3 inserted at the MAT locus, and I don't believe that they directly scored noncrossovers—rather, the other events they detected were mostly recombination between MAT repeats. I think that it would be more accurate to say that "higher levels of interhomolog divergence at a budding yeast hotspot causes decreased crossover and increases the frequency of other events". Their interpretation, which I believe still holds, is not that divergence shifted the balance between crossovers and noncrossovers, but that mismatch repair acting on recombinants (or intermediates) causes secondary events. This is what would be expected from either heteroduplex rejection and/or colliding excision tracts initiating secondary events.

Line 180, 318. "DCOs were significantly greater" could be taken to mean that the frequency of DCOs is greater than random, which I don't think is intended here. Perhaps "the distance between COs in DCOs was significantly greater than predicted by a random model" or something less wordy?

Line 596. The statement is not correct. Rockmill et al report COs increasing about 1.4-fold in *sgs1* mutants. Jessop et al. report no significant increase in SK1 but a ~1.5-fold increase in BR. De Muyt et al. report COs at similar levels in wild-type and *sgs1*, as do Oh et al and Zakharyevich et al. Furthermore, the aberrant joint molecules that accumulate during meiosis I prophase do so at all times during meiosis I prophase, not just at late times.

Figure S1 B. This panel is difficult to parse. Suggest paired column plots of DCO distances (violin-type plots that display all points with bars for mean and SD or median and quartiles) for actual and random for each of the crosses. Could do the same for Figure S9.

Referee #2:

The authors have improved the manuscript and addressed most comments in a satisfactory manner. I still remain with the feeling that the title oversimplifies the findings and might even lead to confusion and that the authors are aiming for a simple bottom line for a complex phenomenon. The strength of their work was to show the complexity. MSH2 is not as well behaved as we could think from the title.

Looking at Figure 4B for example, in the Col x Ler F2 population, in Chromosome 1, MSH2 has no effect in the high diversity pericentric regions, in Chromosome 2, there is an anti-recombination effect at the left of the centromere and a pro-recombination effect at the right of the centromere in the pericentric region. In chromosome 3 there is a pro-recombination effect on both sides of the centromere. Chromosome 4 is like chromosome 2 (one side pro one side anti), and in Chromosome 5 there is no difference between MSH2 and *msh2* in the pericentric region. On top of that there is the antirecombination effect of MSH2 in the distal, less polymorphic regions which is as prominent if not more than the pro recombination effect.

In other words, MSH2 can have an antirecombination role, pro-recombination role or no effect on recombination, depending on the chromosome region and the SNPs density in this region, the genetic background (cross analyzed), the juxtaposition of different regions.

BTW, I could not find explanations for the color code representing the SNPs density (blue, green and mix of purple and green in Col x CLC).

Referee #3:

This revised version of the manuscript has been considerably improved. Most of my concerns have been addressed except for the MSH2 IF study. Since it does not bring much to the study, I suggest that the authors simply remove this part of the results. Otherwise they need to provide more robust analysis both for co-localisation and staging.

Another point is that they appear to have used data from somatic tissues to compare methylation and CO profiles. Now that methylation data of the meiocytes are available (Walker et al. 2018), they should use these ones which are much more relevant.

Thank you for your letter concerning our manuscript EMBOJ-2020-104858R. We have responded to your comments below, which are highlighted in bold, with our response in normal type.

Thank you for submitting your revised manuscript to The EMBO Journal. All three original reviewers have now looked at it again, and found their key criticisms generally satisfactorily addressed. They still retain several minor/specific concerns, as you will see from the comments below, which I would ask you to respond to and/or incorporate during a final round of revision.

We are glad the reviewers were generally satisfied by our response and we have attended to the remaining points, as explained below.

In particular, referee 2 still finds the title/conclusions somewhat too categorically stated, so please consider whether there may be good ways to reconcile this and rephrase in a more qualified manner.

We have changed the title as follows: 'MSH2 shapes the meiotic crossover landscape in relation to interhomolog polymorphism in Arabidopsis'. We hope this reconciles the title with the reviewer's perspective.

Furthermore, as referee 3 is still concerned about the MSH2 immunofluorescence analyses, it should be helpful to include some additional image analysis (as discussed on page 18 of your response letter) if you would like to keep these data in the (main) manuscript.

The Higgins laboratory has reopened since we submitted our first revision and we now include the requested additional analysis.

Specifically, we have quantified MSH2 and MSH4 immunostained foci and measured overlap in male meiocytes. We observed an average of 186 MSH2 foci per cell, of which 131 (74%) overlapped MSH4 foci. As a control, MSH2 images were then rotated 180 degrees relative to the MSH4 image and overlap re-quantified. Following rotation, 65 MSH2 foci on average overlapped MSH4 foci (36%). The observed overlap values were significantly greater than those obtained following rotation (t-test $P=2.25 \times 10^{-3}$). We have added description of this additional analysis to the relevant section of the manuscript.

Regarding referee 3's point about methylation data comparison with somatic vs meiotic tissues, I understand that this concern had not been raised initially, so would not insist on including this - unless of course if it should be straightforward to add. In any case, please again add a brief response letter to the re-reviews upon final resubmission.

We have added an additional plot to Figure S2 including the Walker et al. meiotic DNA methylation data, for comparison with the somatic dataset already provided. As reported by Walker et al., there are differences in non-CG methylation during meiosis compared to somatic cells, yet the centromeric regions remain heavily methylated in both cases.

This supports our point that the regions around the centromeres that are crossover suppressed are also heterochromatic with high levels of DNA methylation.

Referee #1:

Blackwell and co-authors have addressed my concerns, and I think that their paper is ready to publish. I have the following comments, which can be addressed without a need for re-review.

Line 83-85. Actually, Borts and Haber looked at an artificial hotspot containing pBR322 and URA3 inserted at the MAT locus, and I don't believe that they directly scored noncrossovers-rather, the other events they detected were mostly recombination between MAT repeats. I think that it would be more accurate to say that "higher levels of interhomolog divergence at a budding yeast hotspot causes decreased crossover and increases the frequency of other events". Their interpretation, which I believe still holds, is not that divergence shifted the balance between crossovers and noncrossovers, but that mismatch repair acting on recombinants (or intermediates) causes secondary events. This is what would be expected from either heteroduplex rejection and/or colliding excision tracts initiating secondary events.

We have made the suggested wording change in the Introduction.

Line 180, 318. "DCOs were significantly greater" could be taken to mean that the frequency of DCOs is greater than random, which I don't think is intended here. Perhaps "the distance between COs in DCOs was significantly greater than predicted by a random model" or something less wordy?

We have changed the wording here as follows; 'This showed that the distances between observed DCOs were significantly greater than the random distances, in all populations (all $P < 0.005$) (Appendix Fig. S1).'

Line 596. The is statement is not correct. Rockmill et al report COs increasing about 1.4-fold in *sgs1* mutants. Jessop et al. report no significant increase in SK1 but a ~1.5-fold increase in BR. De Muyt et al. report COs at similar levels in wild-type and *sgs1*, as do Oh et al and Zakharyevich et al. Furthermore, the aberrant joint molecules that accumulate during meiosis I prophase do so at all times during meiosis I prophase, not just at late times.

We have changed the wording of the Discussion as follows, 'In contrast, budding yeast *sgs1* mutants accumulate aberrant joint molecules during meiosis and crossovers are either reduced or unchanged (Zakharyevich *et al*, 2012; Oh *et al*, 2007; Rockmill *et al*, 2003; De Muyt *et al*, 2012; Jessop *et al*, 2006).'

Figure S1 B. This panel is difficult to parse. Suggest paired column plots of DCO distances (violin-type plots that display all points with bars for mean and SD or median and quartiles) for actual and random for each of the crosses. Could do the same for Figure S9.

We have explored various ways of visualizing these data. However, we feel that our current plots present the data clearly and allow comparison of observed DCO distances with the random expectation. Therefore, we would like to keep this form of plotting in the final publication.

Referee #2:

The authors have improved the manuscript and addressed most comments in a satisfactory manner. I still remain with the feeling that the title oversimplifies the findings and might even lead to confusion and that the authors are aiming for a simple bottom line for a complex phenomenon. The strength of their work was to show the complexity. MSH2 is not as well behaved as we could think from the title.

We have changed the title as follows:

'MSH2 shapes the meiotic crossover landscape in relation to interhomolog polymorphism in Arabidopsis'.

Looking at Figure 4B for example, in the Col x Ler F2 population, in Chromosome 1, MSH2 has no effect in the high diversity pericentric regions, in Chromosome 2, there is an anti-recombination effect at the left of the centromere and a pro-recombination effect at the right of the centromere in the pericentric region. In chromosome 3 there is a pro-recombination effect on both sides of the centromere. Chromosome 4 is like chromosome 2 (one side pro one side anti), and in Chromosome 5 there is no difference between MSH2 and msh2 in the pericentric region. On top of that there is the antirecombination effect of MSH2 in the distal, less polymorphic regions which is as prominent if not more than the pro recombination effect.

We agree that the relationship between polymorphism and recombination and the effect of MSH2 is complex. We hope that our new title more accurately reflects this complexity.

In other words, MSH2 can have an antirecombination role, pro-recombination role or no effect on recombination, depending on the chromosome region and the SNPs density in this region, the genetic background (cross analyzed), the juxtaposition of different regions.

We have added the following sentence to the Discussion to reflect these important points: 'Hence, it is possible that the effect of MSH2 on crossovers depends on the chromosome region, the level and type of polymorphism, genetic background and the juxtaposition with surrounding regions.'

BTW, I could not find explanations for the color code representing the SNPs density (blue, green and mix of purple and green in Col x CLC).

We have added this information to the legend of Fig. 4B.

Referee #3:

This revised version of the manuscript has been considerably improved. Most of my concerns have been addressed except for the MSH2 IF study. Since it does not bring much to the study, I suggest that the authors simply remove this part of the results. Otherwise they need to provide more robust analysis both for co-localisation and staging.

We would prefer to keep this information in the paper, as it represents the first demonstration of MSH2 association with meiotic chromosomes during prophase I.

We have now also further quantified MSH2 and MSH4 immunostained foci and measured overlap. We observed an average of 186 MSH2 foci per cell, of which 131 (74%) overlapped MSH4 foci. As a control, MSH2 images were then rotated 180 degrees relative to the MSH4 image and the overlap re-quantified. Following rotation, 65 MSH2 foci on average overlapped MSH4 foci (36%). The observed overlap values were significantly greater than those obtained following rotation (t-test $P=2.25 \times 10^{-3}$).

Another point is that they appear to have used data from somatic tissues to compare methylation and CO profiles. Now that methylation data of the meiocytes are available (Walker et al. 2018), they should use these ones which are much more relevant.

We have added an additional plot to Appendix Figure S2 to include the Walker et al. meiotic DNA methylation data, for comparison with the somatic dataset already included. As reported by Walker et al., there are differences in the non-CG methylation between the datasets, yet the centromeric regions remain heavily methylated in both cases. This supports our point that the regions around the centromeres that are crossover suppressed are also heterochromatic with high levels of DNA methylation. We have modified the legend of Appendix Figure S2 to note the different sources of the DNA methylation datasets shown.

Thank you for submitting your final revised manuscript for our consideration. I am pleased to inform you that we have now accepted it for publication in The EMBO Journal.

Corresponding Author Name: Ian Henderson, Piotr Ziolkowski

Journal Submitted to: The EMBO Journal

Manuscript Number: EMBOJ-2020-104858